# Agnostic Continuous-Time Online Learning

**Pramith Devulapalli**[*,1]**, Changlong Wu**[*,1,2,3]**, Ananth Grama**[1,2] **& Wojciech Szpankowski**[1,2,4]
[1]Department of Computer Science, Purdue University
[2]Center for Science of Information and Institute for Physical Artificial Intelligence, Purdue Univ.
[3] University of Arizona    [4]Jagiellonian University
pdevulap@purdue.edu   clwu@arizona.edu   {ayg,szpan}@purdue.edu

## Abstract

We study agnostic online learning from continuous-time data streams, a setting that naturally arises in applications such as environmental monitoring, personalized recommendation, and high-frequency trading. Unlike classical discrete-time models, learners in this setting must interact with a continually evolving data stream while making queries and updating models only at sparse, strategically selected times. We develop a general theoretical framework for learning from both *oblivious* and *adaptive* data streams, which may be noisy and non-stationary. For oblivious streams, we present a black-box reduction to classical online learning that yields a regret bound of $T \cdot R(S)/S$ for any class with discrete-time regret $R(S)$, where $T$ is the time horizon and $S$ is the *query budget*. For adaptive streams, which can evolve in response to learner actions, we design a dynamic query strategy in conjunction with a novel importance weighting scheme that enables unbiased loss estimation. In particular, for hypothesis class $\mathcal{H}$ with a finite Littlestone dimension, we establish a tight regret bound of $\tilde{\Theta}(T \cdot \sqrt{\mathsf{Ldim}(\mathcal{H})/S})$ that holds in both settings. Our results provide the first *quantitative* characterization of agnostic learning in continuous-time online environments with limited interaction.

## 1   Introduction

Online learning from continuous data streams is encountered in many real-world applications, such as environmental sensor networks that sample air quality, recommendation engines that adapt as users click in real time, and high-frequency trading platforms that react rapidly to market transients. Unlike batch or discrete-time online settings, the underlying system state evolves *continually* through time. Consequently, learning algorithms must *query* the system to construct models of the data streams, which may be noisy and non-stationary. Moreover, due to the high cost of retraining online models, both the query and update budgets are often limited.

Given the widespread applications of continuous-time online learning, our objective is to establish its foundations from a learning-theoretic perspective. We are interested in questions such as: How frequently must a learner sample from a continuous stream to maintain accurate predictions? What happens when the stream itself adapts in response to past queries: Can effective learning still take place? And how can we provide strong performance guarantees when no perfect predictor exists?

Our approach to this problem is related to the update-and-deploy framework developed by Devulapalli and Hanneke [7] who recently introduced the setting of continuous-time online learning. Here, the query strategy and the learning algorithm are treated as distinct entities: the query strategy selects continuous-time query points without access to labels, while the learning algorithm updates its parameters only at these discrete query times. During each interval between queries, the predictor remains fixed and processes all incoming data. At each query instant, the learning algorithm observes

---

[*]Equal contribution.

the label, updates the predictor, and redeploys it for the next interval. We note that the theory of continuous-time online learning developed by Devulapalli and Hanneke [7] primarily addresses the *realizable* setting, in which the data stream is assumed to be generated by a function in a given hypothesis class $\mathcal{H}$. However, this assumption may not hold in real-world scenarios, where data is often *noisy* or misspecified.

In this paper, we tackle the more challenging *agnostic* setting [1], where the data stream may be selected arbitrarily and may not align with any function in the hypothesis class. Without assuming the existence of a perfectly consistent hypothesis, our primary objective becomes one of minimizing an appropriate notion of *regret*. Regret measures how much more loss the learner incurs compared to the best fixed predictor from a given hypothesis in hindsight. However, we should emphasize that it is not clear *a priori* how regret should be defined in the continuous-time setting. We introduce two notions of regret, depending on how the data stream is generated, namely, oblivious and adaptive regrets.

**Oblivious Data Streams.** An oblivious data stream in our context is a continuous stream that is arbitrarily chosen in advance before the learning process begins and does not depend on the learner's actions or query times. In this model, regret can be simply defined as the gap between the expected loss incurred by the algorithm when processing the data stream and the minimal expected loss incurred by the best fixed predictor in the hypothesis class on the same data stream.

**Adaptive Data Streams.** A more challenging setting arises when the data stream is *adaptive*; that is, it may change in response to the learner's past queries and deployed predictors. This can occur, for example, when a query made by the learner perturbs the environment. One can model an adaptive data stream as being controlled by an adversary: initially, the adversary commits to a realization of a continuous data stream, but after each query, it may select a new process based on the learner's query times and predictor history. In this case, we define regret as the difference between the expected loss of the algorithm and the expected loss of the best fixed predictor in the hypothesis class, evaluated on the same data stream generated *jointly* by the adversary and the learning algorithm. This notion is analogous to regret formulations used in the bandit learning literature [4].

## 1.1 Main Contributions

We develop *generic* algorithmic frameworks for *agnostic* continuous-time online learning under both oblivious and adaptive data streams with *optimal* regret guarantees. We establish fundamental upper and lower bounds on regret that reveal *precise* trade-offs between query budget $S$, time horizon $T$, and complexity of the hypothesis class $\mathcal{H}$. Together, these results establish a comprehensive theoretical foundation for agnostic continuous-time online learning.

Our main results can be summarized as follows:

1. We show that the regret under *oblivious* data stream can be reduced to *classical* discrete-time online regret via a *black-box* reduction. This is achieved by partitioning the time horizon into $S$ epochs, each of length $\Delta := T/S$, and sampling one query *uniformly* within each epoch. Our main technical insight is that the loss incurred by any predictor on the queried example forms an *unbiased* estimate of the continuous-time risk over the entire epoch. We prove that for any hypothesis class admitting a classical discrete-time regret bound $R(S)$, the corresponding continuous-time regret scales as $T \cdot R(S)/S$; see Theorem 1.

2. For adaptive processes, we show in Proposition 1 that a simple uniform query strategy with fixed epochs *does not* admit sublinear regret. This was resolved by employing a query strategy that samples from *dynamic* (random) epochs as in [7]. However, such a strategy no longer yields an unbiased estimate due to the complex randomness of epoch boundaries. Our main technical innovation is a novel *weighting* scheme that transforms the queried loss into an *unbiased* estimate, even in the presence of random epochs—an idea we believe is of independent interest. More importantly, our weighting scheme is realized *algorithmically* (cf. Algorithm 3), not just in the analysis. Equipped with these techniques, we show in Theorem 2 that adaptive continuous-time regret for any class $\mathcal{H}$ of finite Littlestone dimension scales as $\tilde{O}(T \cdot \sqrt{\mathsf{Ldim}(\mathcal{H})/S})$.

3. Finally, we provide lower bounds that *match* our upper bounds. For finite Littlestone dimensional class $\mathcal{H}$, we show in Theorem 3 that a lower bound of $\Omega(T \cdot \sqrt{\mathsf{Ldim}(\mathcal{H})/S})$ is necessary for *any* algorithm, under *both* oblivious and adaptive data streams. This matches our upper bounds in all relevant parameters: $T$, $S$, and $\mathsf{Ldim}(\mathcal{H})$. Notably, it demonstrates that even though the optimal

query strategies differ between oblivious and adaptive settings, their regrets are of the same order. Our proof follows from novel constructions of *hard* data processes; see Appendix D.

## 1.2 Related Work

Online learning in discrete time has been extensively studied in both the realizable and agnostic settings [12, 4, 16]. A foundational characterization of agnostic learnability for binary-valued hypothesis classes was established by Ben-David et al. [1] via the Littlestone dimension and was subsequently extended to multiclass and real-valued prediction settings [14, 5, 8]. The study of query-efficient and selective sampling models—where the learner decides when to observe *labels*—has provided deep insights into label complexity and active learning trade-offs in both online and batch frameworks [2, 6, 9]. Devulapalli and Hanneke [7] recently initiated the study of continuous-time online learning where they designed the update-and-deploy framework which is a protocol governing how a learning algorithm interacts with a continuous data stream. This setting departs from classical models by requiring the learner to query both features and labels in a continuous-time environment. Although their work establishes a theory for the realizable setting, the agnostic case, particularly its fundamental limits—remains largely unexplored. We address this gap by providing the first *quantitative* characterization of regret in the continuous-time agnostic setting. Our approach introduces several technical innovations, including importance-weighted unbiased loss estimators and the novel applications of Khintchine-type inequalities to derive sharp lower bounds in both the oblivious and adaptive settings. Additionally, our setting is related to problems in nonparametric filtering and online learning with delayed or partial feedback [13, 4].

## 2 Problem Setup

Let $\mathcal{X}$ be an instance space, $\mathcal{Y}$ be a label space, and $\mathcal{H} \subset \mathcal{Y}^{\mathcal{X}}$ be a hypothesis class. A continuous-time data process over $\mathcal{X} \times \mathcal{Y}$ is defined as a collection of joint random variables $\mathbf{Z} := \{(X_t, Y_t)\}_{t \in \mathbb{R}}$, where $Z_t := (X_t, Y_t) \in \mathcal{X} \times \mathcal{Y}$ denotes the random sample at time $t \in \mathbb{R}$.

We consider the update-and-deploy framework from Devulapalli and Hanneke [7] as described in Protocol 1. The learner begins with an initial predictor $\hat{h}_0 \in \mathcal{Y}^{\mathcal{X}}$ and selects the first query time $t_1 \in \mathbb{R}^+$. The predictor $\hat{h}_0$ is used to make predictions on instances $X_t$ for all $t < t_1$, *without* revealing the corresponding sample $(X_t, Y_t)$ to the learner. At query time $t_1$, the environment (or Nature) reveals the labeled example $(X_{t_1}, Y_{t_1})$, after which the learner updates the predictor to $\hat{h}_1$ and selects the next query time $t_2$. This process repeats until a fixed query budget $S \in \mathbb{N}^+$ is exhausted.

---

**Protocol 1** Continuous-Time Online Learning Protocol

---

1: Select initial predictor $\hat{h}_0 \in \mathcal{Y}^{\mathcal{X}}$ and initial query time $t_1 \in \mathbb{R}^+$
2: **for** $s = 1, \dots, S$ **do**
3:    Use $\hat{h}_{s-1}$ to make predictions on $X_t$ for $t \in [t_{s-1}, t_s)$
4:    Query and observe $Z_{t_s} := (X_{t_s}, Y_{t_s})$ at time $t_s$
5:    Choose predictor $\hat{h}_s \in \mathcal{Y}^{\mathcal{X}}$ and the next query time $t_{s+1}$ based on history $\{Z_{t_1}, \dots, Z_{t_s}\}$
6: **end for**

---

Note that the selection of $\hat{h}_s$ and $t_s$ can be *randomized* [2], and that the predictor $\hat{h}_s$ need *not* belong to $\mathcal{H}$ (i.e., is improper). For clarity of exposition, we also assume that the learning process occurs within a finite time horizon $T \in \mathbb{R}^+$. In this case, we also assume $t_0 = 0$ and $t_S = T$ by convention.

Let $\mathcal{A}$ be any continuous-time online learning algorithm, and let $\mathbf{Z} := \{(X_t, Y_t)\}_{t \in \mathbb{R}}$ denote any data process that may depend on $\mathcal{A}$. We define the *risk* of the algorithm as:

$$\mathsf{risk}_T(\mathcal{A}, \mathbf{Z}) := \mathbb{E}\left[\sum_{s=1}^{S} \int_{t_{s-1}}^{t_s} \ell(\hat{h}_{s-1}(X_t), Y_t) \, dt\right], \tag{1}$$

where $\ell : \mathcal{Y} \times \mathcal{Y} \to [0, 1]$ is a bounded loss function, and the expectation is taken over the randomness in $\mathbf{Z}$ and the internal randomness of $\mathcal{A}$. For simplicity, we assume that the function

---

[2]One may also assume that the predictor $\hat{h}_s$ itself is randomized.

$f(t) := \ell(\hat{h}_{s-1}(X_t), Y_t)$ is measurable for any sample path. Similarly, for any (static) hypothesis $h \in \mathcal{H}$, we define the $\mathsf{risk}_T(h, \mathbf{Z})$ by replacing the $\hat{h}_{s-1}$ with $h$ in (1).

For any algorithm $\mathcal{A}$, hypothesis class $\mathcal{H}$ and data process $\mathbf{Z}$, *regret* is defined as:

$$\mathsf{regret}_T(\mathcal{A}, \mathcal{H}, \mathbf{Z}) := \mathsf{risk}_T(\mathcal{A}, \mathbf{Z}) - \inf_{h \in \mathcal{H}} \mathsf{risk}_T(h, \mathbf{Z}). \tag{2}$$

Note that our definition of regret in (2) depends on the data process. To remove this dependency and to quantify the impact of the structural properties of $\mathcal{H}$ on regret, we introduce the following two notions of *minimax* regret for the *worst-case* data processes.

**Oblivious Minimax Regret.** In this setting, we assume that the data process $\mathbf{Z}$ is selected arbitrarily but *independent* (i.e., oblivious) of the internal randomness of $\mathcal{A}$. The *oblivious minimax regret* for a hypothesis class $\mathcal{H}$ is then defined as:

$$\mathsf{regret}_T^o(\mathcal{H}, S) := \inf_{\mathcal{A}} \sup_{\mathbf{Z}} \mathsf{regret}_T(\mathcal{A}, \mathcal{H}, \mathbf{Z}), \tag{3}$$

where the infimum is taken over all learning algorithms $\mathcal{A}$ that make at most $S$ queries and the supremum is taken over all data-generating processes $\mathbf{Z}$ that may depend on the algorithm's structure but *not* on its internal random choices (such as the realized $\hat{h}_s$'s and $t_s$'s).

**Adaptive Minimax Regret.** A more challenging setting arises when the data process $\mathbf{Z}$ can *adapt* to the learner's behavior. In this case, Nature may select an initial process $\mathbf{Z}^{(1)}$ and, after observing the $s$'th query at time $t_s$, switch to a new process $\mathbf{Z}^{(s+1)}$ based on the full interaction history up to that point (i.e., the realized $\{(t_i, \hat{h}_i)\}_{i \leq s}$ but *not* the next query time $t_{s+1}$).

Let $\mathcal{A}$ be a learning algorithm and $\Phi$ be an adaptive strategy for Nature, we define the adaptive risk

$$\mathsf{risk}_T^a(\mathcal{A}, \Phi) := \mathbb{E}\left[\sum_{s=1}^{S} \int_{t_{s-1}}^{t_s} \ell(\hat{h}_{s-1}(X_t^{(s)}), Y_t^{(s)}) \mathrm{d}t\right], \tag{4}$$

where the expectation is over the joint distribution of $\{\hat{h}_{s-1}\}_{s \in [S]}$, $\{t_s\}_{s \in [S]}$ and $\{\mathbf{Z}^{(s)}\}_{s \in [S]}$ induced by $\mathcal{A}$ and $\Phi$, and we denote $\mathbf{Z}^{(s)} := \{(X_t^{(s)}, Y_t^{(s)})\}_{t \in \mathbb{R}}$. We can also define the adaptive risk for any (static) hypothesis $h \in \mathcal{H}$ by replacing the $\hat{h}_{s-1}$ with $h$ in (4), denoted by: $\mathsf{risk}_T^a(h, \mathcal{A}, \Phi)$.

Then, the adaptive minimax *regret* is defined as:

$$\mathsf{regret}_T^a(\mathcal{H}, S) := \inf_{\mathcal{A}} \sup_{\Phi} \left[\mathsf{risk}_T^a(\mathcal{A}, \Phi) - \inf_{h \in \mathcal{H}} \mathsf{risk}_T^a(h, \mathcal{A}, \Phi)\right], \tag{5}$$

where $\mathcal{A}$ runs over all algorithms with $\leq S$ queries and $\Phi$ runs over all adaptive strategies for Nature.

## 3 Main Results

We start with a simple query strategy shown in Algorithm 1 for *oblivious* data processes. The core idea is to partition the time horizon into $S$ epochs, each of length $\Delta := T/S$. Within each epoch, we select a query time uniformly at random and observe a sample. These queried samples are then fed into a classical online learning algorithm, which produces the predictor $\hat{h}_s$ used for the *next* epoch. [3]

The rationale behind this approach is that since the sample is uniformly queried from an interval, the loss incurred by $\hat{h}_{s-1}$ on the *queried* sample is an *unbiased* estimate of the actual (continuous) risk for that interval. Therefore, any regret guarantee achieved by the expert algorithm $\mathcal{B}$ on the queried samples can be translated to the regret of our continuous-time online algorithm (Algorithm 1) in a *black-box* fashion. This observation is formally stated in the following lemma:

---

[3]Note that Algorithm 1 slightly deviates from the general protocol described in Protocol 1 by delaying the update of $\hat{h}_s$ until the start of the next epoch (not at query time). This can be fitted by adding "dummy" queries at the time points $\{s\Delta : s \in [S]\}$, and will only increase the query budget by a factor of 2.

---

**Algorithm 1** Uniform Random Query with Fixed Epochs

---

1: Partition the time horizon into blocks of length $\Delta := T/S$ and select $\hat{h}_0$ arbitrarily
2: **for** $s = 1, 2, \ldots, S$ **do**
3:     Deploy predictor $\hat{h}_{s-1}$ for the *entire* epoch $[(s-1)\Delta, s\Delta)$
4:     Sample $t_s$ uniformly from $[(s-1)\Delta, s\Delta)$.
5:     Query and observe $Z_{t_s} := (X_{t_s}, Y_{t_s})$ at step $t_s$.
6:     Use any expert algorithm $\mathcal{B}$ to produce $\hat{h}_s$ with data $\{Z_{t_1}, \cdots, Z_{t_s}\}$
7: **end for**

---

**Lemma 1.** *Let $\mathbf{Z}$ be any (oblivious) data process, and $\{(t_s, \hat{h}_{s-1}, Z_{t_s})\}_{s \in [S]}$ be the (random) selections generated by Algorithm 1. Then, for any $s \in [S]$ and any (random) predictor $h_{s-1} \in \mathcal{Y}^{\mathcal{X}}$ that depends only on $\{(t_i, \hat{h}_{i-1}, Z_{t_i})\}_{i \in [s-1]}$, we have:*

$$\mathbb{E}\left[\ell(h_{s-1}(X_{t_s}), Y_{t_s})\right] = \frac{1}{\Delta}\mathbb{E}\left[\int_{(s-1)\Delta}^{s\Delta} \ell(h_{s-1}(X_t), Y_t)\mathrm{d}t\right],$$

*where the expectation is over all randomness involved.*

*Proof.* By the law of total probability, we have

$$\mathbb{E}\left[\ell(h_{s-1}(X_{t_s}), Y_{t_s})\right] = \mathbb{E}\left[\mathbb{E}_{t_s}\left[\ell(h_{s-1}(X_{t_s}), Y_{t_s}) \mid \{(t_i, \hat{h}_{i-1})\}_{i \in [s-1]}, \mathbf{Z}\right]\right]$$

$$= \mathbb{E}\left[\frac{1}{\Delta}\int_{(s-1)\Delta}^{s\Delta} \ell(h_{s-1}(X_t), Y_t)\mathrm{d}t\right],$$

where we used the fact that conditioning on $\{(t_i, \hat{h}_{i-1})\}_{i \in [s-1]}$ and $\mathbf{Z}$, the index $t_s$ is independent of $h_{s-1}$ and is uniform over $[(s-1)\Delta, s\Delta)$ (we then express the expectation on $t_s$ as an integral).  $\square$

Note that Lemma 1 holds for any predictor $h_{s-1}$ that satisfies the stated condition (including a *static* predictor), not just for the produced predictor $\hat{h}_{s-1}$. We now introduce the *classic* regret as:

$$\tilde{\mathrm{regret}}_S(\mathcal{B}, \mathcal{H}) := \sup_{Z_{t_1}, \cdots, Z_{t_S}} \left\{\sum_{s=1}^{S} \mathbb{E}\left[\ell(\hat{h}_{s-1}(X_{t_s}), Y_{t_s})\right] - \inf_{h \in \mathcal{H}} \sum_{s=1}^{S} \ell(h(X_{t_s}), Y_{t_s})\right\}, \quad (6)$$

where the expectation is over the internal randomness of $\mathcal{B}$ that produces $\hat{h}_{s-1}$.

**Theorem 1.** *Let $\mathcal{H}$ be any hypothesis class that admits an expert algorithm $\mathcal{B}$ with $\tilde{\mathrm{regret}}_S(\mathcal{B}, \mathcal{H}) \leq R(S)$ for some function $R : \mathbb{N} \to \mathbb{R}$. Then, the oblivious minimax regret of $\mathcal{H}$ satisfies*

$$\mathrm{regret}_T^o(\mathcal{H}, S) \leq \Delta \cdot R(S) = \frac{T \cdot R(S)}{S}.$$

*Moreover, this upper bound is achieved by Algorithm 1 using the expert algorithm $\mathcal{B}$.*

*Proof.* Let $\mathbf{Z}$ be any fixed data process, and $\{t_s, \hat{h}_{s-1}, Z_{t_s}\}_{s \in [S]}$ be the selections made by Algorithm 1 using expert algorithm $\mathcal{B}$. Invoking Lemma 1 and the linearity of expectation, we have

$$\mathrm{risk}_T(\mathcal{A}, \mathbf{Z}) \stackrel{\mathrm{def}}{=} \mathbb{E}\left[\sum_{s=1}^{S} \int_{(s-1)\Delta}^{s\Delta} \ell(\hat{h}_{s-1}(X_t), Y_t)\,\mathrm{d}t\right] = \Delta \cdot \mathbb{E}\left[\sum_{s=0}^{S-1} \ell(\hat{h}_{s-1}(X_{t_s}), Y_{t_s})\right],$$

where the expectation is over all randomness involved. Similarly, for any (static) $h \in \mathcal{H}$, we have $\mathrm{risk}_T(h, \mathbf{Z}) = \Delta \cdot \mathbb{E}\left[\sum_{s=1}^{S} \ell(h(X_{t_s}), Y_{t_s})\right]$. Therefore,

$$\frac{1}{\Delta}\mathrm{regret}_T(\mathcal{A}, \mathcal{H}, \mathbf{Z}) = \mathbb{E}\left[\sum_{s=1}^{S} \ell(\hat{h}_{s-1}(X_{t_s}), Y_{t_s})\right] - \inf_{h \in \mathcal{H}} \mathbb{E}\left[\sum_{s=1}^{S} \ell(h(X_{t_s}), Y_{t_s})\right]$$

$$\leq \mathbb{E}\left[\sum_{s=1}^{S}\ell(\hat{h}_{s-1}(X_{t_s}),Y_{t_s}) - \inf_{h\in\mathcal{H}}\sum_{s=1}^{S}\ell(h(X_{t_s}),Y_{t_s})\right] \leq R(S),$$

where the first inequality follows by $\inf \mathbb{E} \geq \mathbb{E}\inf$ and linearity of expectation. This completes the proof, and the final identity follows by $\Delta := T/S$. $\qquad\square$

Instantiating Theorem 1 to specific hypothesis classes, we have:

**Corollary 1.** *Let* $\mathcal{Y} = [0,1]$ *and* $\mathcal{H} \subset \{0,1\}^{\mathcal{X}}$ *be a binary-valued class of Littlestone dimension* $\mathsf{Ldim}(\mathcal{H})$. *Then, for absolute loss* $\ell(y,y') := |y-y'|$, *the* oblivious *minimax regret of* $\mathcal{H}$ *satisfies*

$$\mathsf{regret}_T^o(\mathcal{H},S) \leq O\left(\frac{T\cdot\sqrt{\mathsf{Ldim}(\mathcal{H})\cdot\log S}}{\sqrt{S}}\right).$$

*Proof.* By [1], class $\mathcal{H}$ admits an expert algorithm $\mathcal{B}$ with *classic* regret upper bounded by $R(S) \leq O(\sqrt{S\log S \cdot \mathsf{Ldim}(\mathcal{H})})$ under the *expected* misclassification loss. The result then follows from Theorem 1 by plugging in $R(S)$ and noting that the *expected* misclassification loss can be interpreted as the absolute loss [4, Chapter 8]. $\qquad\square$

Observe that Corollary 1 provides the *precise* trade-off between the query budget $S$, time horizon $T$ and the resulting regret. For instance, it shows that if the query budget grows linearly as $S = \Omega(T)$, then the regret is upper bounded by $\tilde{O}(\sqrt{T\cdot\mathsf{Ldim}(\mathcal{H})})$. On the other hand, if the budget is sublinear as $T^{\alpha}$ for some $\alpha < 1$, then the regret grows as $\tilde{O}(T^{1-\frac{\alpha}{2}}\cdot\sqrt{\mathsf{Ldim}(\mathcal{H})})$. We will show in Section 3.2 that this regret is *tight* w.r.t. all parameters $T$, $S$ and $\mathsf{Ldim}(\mathcal{H})$.

**Remark 1.** *Note that Theorem 1 provides a* black-box *reduction from any achievable classical discrete-time online regret to the (oblivious) continuous-time regret. For instance, for real-valued class* $\mathcal{H} \subset [0,1]^{\mathcal{X}}$ *with sequential-fat-shattering dimension of order* $\alpha^{-p}$, *we have* $R(S) \leq \tilde{O}(S^{\frac{p-1}{p}})$ *under absolute loss [14]. This leads to the continuous-time risk:* $\mathsf{regret}_T^o(\mathcal{H},S) \leq \tilde{O}(T\cdot S^{-1/p})$. *Furthermore, Algorithm 1 is computationally efficient, provided that the expert algorithm* $\mathcal{B}$ *is efficient (such as the oracle-efficient algorithms in [11, 3, 17]).*

### 3.1 Adaptive Minimax Regret

One might observe that the key ingredient in the proof of Lemma 1 is the *independence* of the query time $t_s$ from the data process $\mathbf{Z}$. This independence allows us to obtain an unbiased estimate of the continuous-time risk over the epoch $[(s-1)\Delta,\, s\Delta)$, even if only a single sample is queried and the process is selected in a completely arbitrary manner.

Unfortunately, this property does not carry over to an *adaptively* selected data process that may depend on the query times. This is demonstrated formally in the following proposition.

**Proposition 1.** *For any hypothesis class* $\mathcal{H} \subset \{0,1\}^{\mathcal{X}}$ *that contains two* $h_1, h_2 \in \mathcal{H}$ *and* $\mathbf{x}_1, \mathbf{x}_2 \in \mathcal{X}$ *such that* $h_1(\mathbf{x}_1) = h_2(\mathbf{x}_1)$ *but* $h_1(\mathbf{x}_2) \neq h_2(\mathbf{x}_2)$, *we can construct an* adaptive *process, such that Algorithm 1 with any* $\mathcal{B}$ *and* $S$ *incurs regret lower bounded by* $\Omega(T)$ *under absolute loss.*

*Proof.* Let $B$ be uniformly sampled from $\{0,1\}$. At each epoch $s$, we set $X_t := \mathbf{x}_1$, $Y_t := h_1(\mathbf{x}_1)$ for $t \in [(s-1)\Delta, t_s]$, and switch to $X_t := \mathbf{x}_2$, $Y_t := B$ for $t \in (t_s, s\Delta)$. Note that this data process depends on the (random) query times $t_s$'s and is *realizable* w.r.t. $\mathcal{H}$; that is, either $h_1$ or $h_2$ incurs 0 risk. Since the queried example is always $(\mathbf{x}_1, h_1(\mathbf{x}_1))$, which is *independent* of $B$. Therefore, the *expected* risk is lower bounded by the risk of $\hat{h}_{s-1}$'s on the intervals $(t_s, s\Delta)$ as:

$$\mathbb{E}_B\mathbb{E}\left[\sum_{s=1}^{S}\mathbb{1}\{\hat{h}_{s-1}(\mathbf{x}_2)\neq B\}\cdot(s\Delta - t_s)\right] = \mathbb{E}\left[\sum_{s=1}^{S}\mathbb{E}_B[\mathbb{1}\{\hat{h}_{s-1}(\mathbf{x}_2)\neq B\}]\cdot(s\Delta - t_s)\right]$$

$$= \mathbb{E}\left[\sum_{s=1}^{S}\frac{1}{2}(s\Delta - t_s)\right] = \frac{\Delta\cdot S}{4} = \frac{T}{4},$$

where the penultimate identity follows by $\mathbb{E}[(s\Delta - t_s)] = \frac{\Delta}{2}$. Therefore, there must *exist* some $B \in \{0,1\}$ such that Algorithm 1 incurs $T/4$ regret for our adaptive process constructed above. $\qquad\square$

Proposition 1 implies that for any class $\mathcal{H} \subset \{0,1\}^{\mathcal{X}}$ with Littlestone dimension at least 2 (which satisfies the stated condition), Algorithm 1 incurs *adaptive* regret $\geq \frac{T}{4}$. This effectively rules out almost all hypothesis classes of interest. Note that the main reason our adaptive process breaks the arguments in the proof of Theorem 1 is that the loss at the query example $t_s$ is no longer an unbiased estimate of the continuous-time risk over the entire epoch $[(s-1)\Delta, s\Delta)$.

To address this issue, we employ a query strategy adapted from [7], by sampling the query time $t_s$ uniformly from the *dynamic* epoch $[t_{s-1}, t_{s-1} + \Delta)$, as described in Algorithm 2. However, since the data process $\mathbf{Z}^{(s+1)}$ after the query time $t_s$ may *change* under an adaptive adversary, an analog of Lemma 1 no longer holds. Our main idea is a novel *weighting scheme* on the queried losses (see Algorithm 3), which serves as a substitute *unbiased* estimator for the continuous-time risks. This is formalized in Lemma 2 below and constitutes the main technical innovation of this section.

---

**Algorithm 2** Uniform Random Query with Dynamic Epochs

---

1: Select the initial predictor $\hat{h}_0 \in \mathcal{Y}^{\mathcal{X}}$ arbitrarily and sample $t_1$ uniform over $[0, \Delta)$.
2: **for** $s = 1, 2, \ldots, S$ **do**
3:     Use $\hat{h}_{s-1}$ to make predictions on $X_t$ for $t \in [t_{s-1}, t_s)$
4:     Query and observe $Z_{t_s}^{(s)} := (X_{t_s}^{(s)}, Y_{t_s}^{(s)})$ at time $t_s$
5:     Use Weighted-EWA (Algorithm 3) to produce $\hat{h}_s$ with data $\{(t_i, Z_{t_i}^{(i)})\}_{i \leq s}$
6:     Sample $t_{s+1}$ uniformly from the *dynamic* epoch $[t_s, t_s + \Delta)$
7: **end for**

---

---

**Algorithm 3** Weighted-EWA

---

1: Let $\mathcal{H} := \{h_1, \cdots, h_k\}$ be finite of size $K$

2: Set initial weight $\mathbf{w}^0 = (1, \cdots, 1) \in [0,1]^K$ and the learning rate $\eta = \sqrt{\frac{\log K}{S}}$

3: **for** $s = 1, 2, \ldots, S$ **do**

4:     Retrieve data $(t_s, Z_{t_s}^{(s)})$

5:     For all $k \in [K]$, update weight

$$\mathbf{w}^s[k] = \mathbf{w}^{s-1}[k] \cdot e^{-\eta \cdot \left(1 - \frac{t_s - t_{s-1}}{\Delta}\right) \cdot \ell(h_k(X_{t_s}^{(s)}), Y_{t_s}^{(s)})}$$

6:     Produce predictor

$$\hat{h}_s = \frac{\sum_{k=1}^{K} \mathbf{w}^s[k] \cdot h_k}{\sum_{k=1}^{K} \mathbf{w}^s[k]}$$

7: **end for**

---

**Lemma 2.** *Let $\Phi$ be any adaptive strategy and $\{\hat{h}_{s-1}, t_s, \mathbf{Z}^{(s)}\}_{s \in [S]}$ be the realizations generated by interactions between Algorithm 2 and $\Phi$ (see Section 2). Then, for any $s \in [S]$ and any (random) predictor $h_{s-1} \in \mathcal{Y}^{\mathcal{X}}$ that depends only on $\{\hat{h}_{i-1}, t_i, \mathbf{Z}^{(i)}\}_{i \leq s-1}$, we have:*

$$\mathbb{E}\left[\left(1 - \frac{t_s - t_{s-1}}{\Delta}\right) \cdot \ell(h_{s-1}(X_{t_s}^{(s)}), Y_{t_s}^{(s)})\right] = \frac{1}{\Delta} \mathbb{E}\left[\int_{t_{s-1}}^{t_s} \ell(h_{s-1}(X_t^{(s)}), Y_t^{(s)}) \mathrm{d}t\right],$$

*where the expectation is over all randomness involved.*

*Proof.* Conditioning on any realizations of $\{\hat{h}_{i-1}, t_i, \mathbf{Z}^{(i)}\}_{i \leq s-1}$, $h_{s-1}$ and $\mathbf{Z}^{(s)}$, the conditional distribution of $t_s$ remains uniform over $[t_{s-1}, t_{s-1} + \Delta)$. Denote $f(t) := \ell(h_{s-1}(X_t^{(s)}), Y_t^{(s)})$ and $\tau = t_s - t_{s-1}$ for notation convenience. We have

$$\frac{1}{\Delta} \mathbb{E}_{t_s}\left[\int_{t_{s-1}}^{t_s} \ell(h_{s-1}(X_t^{(s)}), Y_t^{(s)}) \mathrm{d}t\right] = \frac{1}{\Delta^2} \int_0^{\Delta} \left(\int_{t_{s-1}}^{t_{s-1}+\tau} f(t) \mathrm{d}t\right) \mathrm{d}\tau$$

$$\stackrel{(\star)}{=} \frac{1}{\Delta^2} \int_{t_{s-1}}^{t_{s-1}+\Delta} \left( \int_{t-t_{s-1}}^{\Delta} 1 \cdot f(t) \mathsf{d}\tau \right) \mathsf{d}t$$

$$= \frac{1}{\Delta^2} \int_{t_{s-1}}^{t_{s-1}+\Delta} (\Delta + t_{s-1} - t) \cdot f(t) \mathsf{d}t$$

$$\stackrel{(\star\star)}{=} \frac{1}{\Delta} \mathbb{E}_{t_s} \left[ (\Delta - \tau) \cdot f(t_s) \right] = \mathbb{E}_{t_s} \left[ \left( 1 - \frac{\tau}{\Delta} \right) \cdot f(t_s) \right],$$

where the key identity $(\star)$ follows by exchanging the order of integrations, and $(\star\star)$ follows by setting $t := t_s$ since $t_s$ is distributed uniformly over $[t_{s-1}, t_{s-1} + \Delta)$. The lemma then follows by taking the expectation over the remaining randomness and by the law of total probability. $\square$

Intuitively, Lemma 2 shows that the *continuous-time* risk of predictor $h_{s-1}$ over the (random) interval $[t_{s-1}, t_s)$ can be estimated by a *weighted* loss of $h_{s-1}$ at the queried sample $Z_{t_s}^{(s)}$. In particular, the weighting factor depends on the query times and discounts those that are far from the previous query time. It is also instructive to observe that the expected loss at the query sample *without* weighting always *overestimates* the continuous-time risk.

The following lemma provides a useful bound on the *discrete-time* (weighted) regret incurred by Algorithm 3 at the query samples. The proof follows from the same regret analysis of the classical EWA algorithm [4, Theorem 2.2] and is deferred to Appendix B for readability.

**Lemma 3.** *Let $\mathcal{H} \subset [0,1]^{\mathcal{X}}$ be any finite class and $\ell : [0,1] \times [0,1] \to [0,1]$ be any loss that is convex in the first argument. Then, Algorithm 3 yields the following regret:*

$$\sup_{\{(t_s, Z_{t_s}^{(s)})\}_{s \in [S]}} \left\{ \sum_{s=1}^{S} \alpha_s \cdot \ell(\hat{h}_{s-1}(X_{t_s}^{(s)}), Y_{t_s}^{(s)}) - \inf_{h \in \mathcal{H}} \sum_{s=1}^{S} \alpha_s \cdot \ell(h(X_{t_s}^{(s)}), Y_{t_s}^{(s)}) \right\} \leq O(\sqrt{S \log |\mathcal{H}|}),$$

*where $\alpha_s := \left( 1 - \frac{t_s - t_{s-1}}{\Delta} \right) \in [0,1]$ and $\{(t_s, Z_{t_s}^{(s)})\}_{s \in [S]}$ run over all possible selections.*

Note that the regret guarantee of Lemma 3 holds for any selection of the data $\{(t_s, Z_{t_s}^{(s)})\}_{s \in [S]}$, including *adaptive* strategies. It remains valid even when the $\alpha_s$'s take different forms.

We are now ready to state our main technical result for this section.

**Lemma 4.** *Let $\mathcal{H} \subset [0,1]^{\mathcal{X}}$ be any finite class and $\ell : [0,1] \times [0,1] \to [0,1]$ be any loss that is convex in the first argument. Then, the* adaptive *minimax regret of $\mathcal{H}$ satisfies:* $\mathsf{regret}_T^a(\mathcal{H}, S) \leq O(T \cdot \sqrt{(\log |\mathcal{H}|)/S})$. *Moreover, this is achieved by Algorithm 2 with $\Delta := \frac{4T}{S}$.*

*Proof.* The proof follows the same argument as Theorem 1, by replacing Lemma 1 with Lemma 2, and invoking Lemma 3 to bound the *weighted* regret on the queried samples. The only remaining step is to select an appropriate $\Delta$ that ensures $t_S \geq T$, since the epochs $[t_{s-1}, t_s)$ are no longer fixed. To this end, by the *multiplicative* Chernoff bound [18, Corollary 2.18], we have:

$$\Pr\left[ t_S = \sum_{s=1}^{S} (t_s - t_{s-1}) < T \right] \leq \exp\left( -\frac{S}{4} \left( 1 - \frac{2T}{S\Delta} \right)^2 \right).$$

Setting $\Delta := 4T/S$, the error probability becomes $\exp(-S/16)$. Therefore, the expected risk contributed by the "bad" event $\{t_S < T\}$ is at most $T \cdot \exp(-S/16)$, which is negligible. $\square$

Using a covering argument similar to that in [1, Lemma 12], we extend Lemma 4 to hypothesis classes with finite Littlestone dimension below. The full proof is deferred to Appendix C.

**Theorem 2.** *Let $\mathcal{H} \subset \{0,1\}^{\mathcal{X}}$ be a binary-valued class of finite Littlestone dimension and $\ell$ be the absolute loss $\ell(y, y') = |y - y'|$. Then, the* adaptive *minimax regret for $\mathcal{H}$ satisfies:*

$$\mathsf{regret}_T^a(\mathcal{H}, S) \leq O\left( \frac{T \cdot \sqrt{\mathsf{Ldim}(\mathcal{H}) \cdot \log S}}{\sqrt{S}} \right).$$

We remark that the use of the EWA algorithm in the proof of Lemma 4 is not essential. In fact, any algorithm that achieves low *weighted* regret, as in Lemma 3, enables a *black-box* reduction. This includes algorithms from the online convex optimization literature [10], such as online gradient descent and the Follow-the-Regularized-Leader (FTRL) family of methods.

## 3.2 Lower Bounds

In the preceding sections, we presented algorithms for both oblivious and adaptive data streams that achieve sublinear regret rates. We now show that no algorithm with a bounded query budget can attain a better rate. Specifically, we establish *matching* lower bounds in both settings, thereby demonstrating that our regret guarantees are *optimal*.

**Theorem 3.** *Let $\mathcal{Y} = [0, 1]$, $\mathcal{H} \subset \{0, 1\}^{\mathcal{X}}$ be any class with Littlestone dimension $\mathsf{Ldim}(\mathcal{H})$ and $\ell$ be the absolute loss $\ell(y, y') = |y - y'|$. Then, for any algorithm $\mathcal{A}$ with query budget S, there exists an oblivious data process $\mathbf{Z}$ such that:*

$$\mathsf{risk}_T^o(\mathcal{A}, \mathbf{Z}) - \inf_{h \in \mathcal{H}} \mathsf{risk}_T^o(h, \mathbf{Z}) \geq \Omega\left(\frac{T\sqrt{\mathsf{Ldim}(\mathcal{H})}}{\sqrt{S}}\right).$$

Note that the lower bound in Theorem 3 holds for *oblivious* data processes, which automatically applies to adaptive data processes. This implies that *both* upper bounds in Corollary 1 and Theorem 2 are *tight* (up to a $\log S$ factor) w.r.t. *all* parameters: $T$, $S$, and $\mathsf{Ldim}(\mathcal{H})$.

To proceed, we introduce the following key inequality (see [4, Lemma A.9]).

**Lemma 5** (Khinchine's Inequality). *Let $\epsilon_1, \cdots, \epsilon_n$ be uniform over $\{-1, +1\}^n$ and $a_1, \cdots, a_n$ be real numbers. Then $\frac{1}{\sqrt{2}} \left(\sum_{i=1}^n a_i^2\right)^{1/2} \leq \mathbb{E}|\sum_{i=1}^n a_i \epsilon_i| \leq \left(\sum_{i=1}^n a_i^2\right)^{1/2}$.*

We present here a simplified proof under *adaptive* data processes and assume that the query strategy is *data-independent*. The stronger result for *oblivious* data processes and general query strategies follows from a more intricate construction; the complete proof is deferred to Appendix D.

*Proof of Theorem 3 (weaker version).* We first consider the case where $\mathcal{H} := \{h_0, h_1\}$ with $h_0(\mathbf{x}) = 0$ and $h_1(\mathbf{x}) = 1$ for some fixed $\mathbf{x}$. Let $\{b_1, \cdots, b_S\} \sim \mathsf{Unif}(\{0, 1\}^S)$, we construct an adaptive data process $\Phi^{(b)}$ as follows. We set the initial data process $Z_t^{(1)} := (\mathbf{x}, b_1)$ for all $t \in \mathbb{R}$. After the $s$ query, we switch to another process with $Z_t^{(s+1)} := (\mathbf{x}, b_{s+1})$ for all $t \in \mathbb{R}$. Denote $T_s := t_s - t_{s-1}$ as the (random) epoch length between the $s$ and $s - 1$ queries. We have

$$\mathbb{E}_b\left[\mathsf{risk}_T^a(\mathcal{A}, \Phi^{(b)})\right] = \mathbb{E}_b\mathbb{E}\left[\sum_{s=1}^S \int_{t_{s-1}}^{t_s} \ell(\hat{h}_{s-1}(X_t^{(s)}), Y_t^{(s)})\mathrm{d}t\right]$$

$$= \mathbb{E}\left[\sum_{s=1}^S \int_{t_{s-1}}^{t_s} \mathbb{E}_{b_s}[|\hat{h}_{s-1}(\mathbf{x}) - b_s|]\right] = \mathbb{E}\left[\sum_{s=1}^S \int_{t_{s-1}}^{t_s} \frac{1}{2}\right] = \frac{T}{2}$$

where we used the independence between $b_s$ and $\{\hat{h}_{s-1}, t_s\}$. For comparator loss, we have:

$$\mathbb{E}_b\left[\inf_{h \in \mathcal{H}} \mathbb{E}\left[\sum_{s=1}^S \int_{t_{s-1}}^{t_s} |h(\mathbf{x}) - b_s|\right]\right] = \mathbb{E}_b\left[\min_{y \in \{0,1\}} \sum_{s=1}^S \mathbb{E}[T_s] \cdot |y - b_s|\right]$$

$$\stackrel{(a)}{=} \frac{T}{2} + \frac{1}{2}\mathbb{E}_\epsilon\left[\min\left\{\sum_{s=1}^S a_s\epsilon_s, -\sum_{s=1}^S a_s\epsilon_s\right\}\right]$$

$$= \frac{T}{2} - \frac{1}{2}\mathbb{E}_\epsilon\left[\left|\sum_{s=1}^S a_s\epsilon_s\right|\right]$$

$$\stackrel{(b)}{\leq} \frac{T}{2} - \frac{1}{2\sqrt{2}}\sqrt{\sum_{s=1}^S a_s^2} \stackrel{(c)}{\leq} \frac{T}{2} - \frac{T}{2\sqrt{2S}}, \tag{7}$$

where $(a)$ follows by changing of variables $\epsilon_s := \frac{1-b_s}{2}$ and setting $a_s := \mathbb{E}[T_s]$; $(b)$ follows by Lemma 5 (lower bound); $(c)$ follows by Cauchy–Schwarz and the fact that $a_1 + \cdots + a_S = T$. By the probabilistic method, there must *exist* some $\Phi^{(b)}$ that attains an expected regret of $\Omega(T/\sqrt{S})$.

For general class $\mathcal{H}$ with Littlestone dimension $d := \mathsf{Ldim}(\mathcal{H})$, we follow an approach similar to that of [1, Lemma 14]. We divide the query budget into $d$ *phases*, each with $S/d$ queries. The adversarial

strategy proceeds as follows: let $\tau$ be a Littlestone tree of $\mathcal{H}$ of depth $d$. We begin with $\mathbf{x}_1$ as the root of $\tau$ and assign the labels using the construction for the simplified case above. Once the learner uses all $S/d$ queries, we choose the next input $\mathbf{x}_2$ as the child of $\mathbf{x}_1$ in $\tau$ whose outgoing edge label minimizes the comparator loss in (7). This process continues until all query budget is exhausted. Let $T_r$ be the length of the time-interval for the $r$'th phase, the total regret is lower bounded by:

$$\sum_{r=1}^{d} \frac{T_r}{2\sqrt{2S/d}} \geq \Omega\left(\frac{T}{\sqrt{S/d}}\right) = \Omega\left(\frac{T\sqrt{d}}{\sqrt{S}}\right).$$

Here, we used the fact that there exists $h \in \mathcal{H}$ whose predictions attain (7) for all phases. $\qquad\square$

**Remark 2.** *Note that step (b) in Eq. (7) implicitly uses the data-independence assumption of the queries (i.e., the $a_s$ is independent of $\epsilon_s$'s). This assumption is removed in our full proof (Appendix D) by constructing an oblivious data process. Furthermore, although our simplified proof here is reminiscent of the classic arguments in [1], its adaptation to the continuous-time setting and the phase-based scheduling constitutes a substantive theoretical advancement.*

## 4  Discussion

Our work focuses primarily on the *worst-case* setting, where the data process is arbitrary and no distributional assumptions are made. In this adversarial regime, we show that our algorithms achieve optimal regret rates, tightly matching lower bounds in both the oblivious and adaptive settings. These results provide a precise characterization of the inherent difficulty of learning from agnostic continuous data streams with limited interaction. However, it is important to note that for specific data processes, improved strategies may be possible by leveraging structural or statistical assumptions. For instance, consider the case where the data process is i.i.d. and is *realizable* w.r.t. a finite hypothesis class. A strategy that queries at uniformly spaced intervals of length $\Delta := T/S$ and updates the predictor using empirical risk minimization (ERM) at each query point incurs total risk of order:

$$\sum_{s=1}^{S} \frac{\Delta}{s} \approx \frac{T\log S}{S},$$

since the risk of ERM with $s$ i.i.d. samples is $O(1/s)$ [15]. In contrast, a naive strategy that queries $S$ examples at the very beginning and freezes the predictor for the remainder of the time horizon incurs a total risk of order $T/S$. Therefore, designing optimal query strategies tailored to specific data-generating processes remains an intriguing direction for future research.

Another modeling choice that may raise concern is our use of integration over time to define risk, which may appear idealized in real-world settings, where predictions are inherently discrete. We emphasize that the integral formulation is primarily a mathematical abstraction used to simplify exposition and analysis. An analogous discrete-time version, where risk is computed as the sum of per-step losses, leads to the same bounds under mild assumptions. In that case, care must be taken to ensure that the number of queries does not exceed the total number of time steps, but all of our arguments and regret guarantees extend naturally with minor modifications.

**Acknowledgments.**    This work is partially supported by the NSF Center for Science of Information (CSoI) Grant CCF-0939370, and also by NSF Grants CCF-2006440 and and CCF-2211423.

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

## A   Empirical Validation

We complement our theoretical results with controlled simulations to illustrate how the proposed update rules behave under various data processes. We implement three algorithms corresponding to our theoretical constructions:

1. **Uniform Epoch:** updates are performed at fixed, evenly spaced intervals.
2. **Dynamic Epoch:** update times are chosen adaptively based on past intervals.
3. **Dynamic + Reweighting:** same as Dynamic, but each update is weighted by

$$\alpha_s = 1 - \frac{t_s - t_{s-1}}{\Delta},$$

to correct post-epoch bias.

All algorithms use the *Exponential Weights Algorithm (EWA)* with absolute loss over a class of threshold experts,
$$\mathcal{H} = \{\, h_\tau(x) = \mathbf{1}\{x \geq \tau\} : \tau \in [0,1] \,\}.$$
We instantiate $K = 201$ uniformly spaced thresholds $\tau_j = \frac{j}{K+1}$, and simulate over $N = 30{,}000$ discrete time steps (corresponding to a continuous horizon of $T = 1$) to approximate the integration in our theoretical formulation. The number of model updates is varied as

$$S \in \{25,\ 40,\ 64,\ 80,\ 100,\ 128,\ 160,\ 200,\ 256,\ 320,\ 400,\ 512,\ 640,\ 800,\ 1024,\ 1280\}.$$

Each configuration is averaged over five independent random seeds to reduce stochastic variance.

**Data processes.**   In all environments, the features are generated independently as $x_t \sim \mathrm{Unif}[0,1]$. The environments differ only in how the labels $y_t$ are generated as (possibly adversarial) functions of $x_t$ and the query times $\{t_s\}$. We consider one stochastic and three adversarial constructions:

- **Oblivious Drift:** Let $t \mapsto \theta(t) = 0.5 + 0.3\sin(4\pi t/T)$ denote a drifting threshold completing two full cycles over the horizon $[0,T]$. At each discrete tick $t$, the label is assigned as

$$y_t = \mathbf{1}\{x_t \geq \theta(t)\},$$

  followed by 5% independent label flips ($y_t \leftarrow 1 - y_t$ with probability 0.05). This produces a smooth periodic drift that is entirely oblivious to the learner's updates.

- **Adversarial A1 (Bit-after-Query):** Labels alternate over time between two regimes, with switches exactly at the learner's query times. One regime uses the informative threshold $y_t = \mathbf{1}\{x_t \geq 0.5\}$; the other fixes labels to a constant random bit $B_s \sim \mathrm{Bernoulli}(1/2)$.

- **Adversarial A2 (Post-Query Shift):** Labels alternate over time between two threshold regimes, with switches exactly at the learner's query times. One regime follows the balanced rule $y_t = \mathbf{1}\{x_t \geq 0.5\}$; the other adopts a skewed rule $y_t = \mathbf{1}\{x_t \geq 0.8\}$, creating abrupt post-query shifts that make the data imbalanced and harder to predict.

- **Adversarial A3 (Refresh-Random-Bits):** After each query, the label switches to a freshly sampled random bit $B_s \sim \mathrm{Bernoulli}(1/2)$, which is held fixed until the next query.

**Evaluation.**   We measure the mean regret

$$R(S) := \frac{1}{N}\sum_{t=1}^{N} |\hat{y}_t - y_t| - \min_j \frac{1}{N}\sum_{t=1}^{N} |h_{\tau_j}(x_t) - y_t|,$$

and report mean $\pm$ std across five seeds. For visual reference, we also fit the curve $C/\sqrt{S}$ in log-space.

**Results.**   Figure 1 plots the average regret versus $S$ (log–log scale), together with a fitted $C/\sqrt{S}$ reference line. Across all settings, the regret decays approximately as $O(1/\sqrt{S})$, consistent with our theoretical prediction. Moreover, the deviations around the mean are of small order, indicating that the behavior holds not only in expectation but is also robust across runs.

- In the **oblivious** case, all three algorithms achieve the expected scaling, differing only by constants.

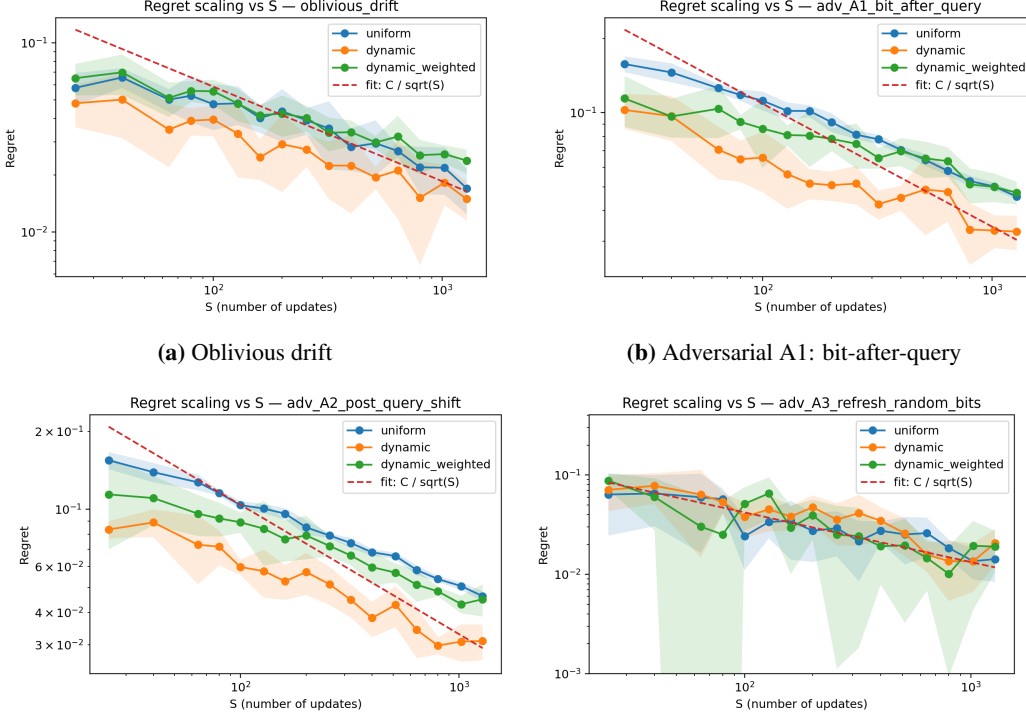

**(a)** Oblivious drift

**(b)** Adversarial A1: bit-after-query

**(c)** Adversarial A2: post-query shift

**(d)** Adversarial A3: refresh random-bits

**Figure 1:** Regret scaling versus number of updates $S$ (log-log scale) under different query strategy and data processes. The dashed red line shows the fitted $C/\sqrt{S}$ reference.

- Under **A1** and **A2**, the *dynamic* schedule outperforms uniform updates, confirming the benefit of adaptive query placement.

- Under **A3**, the dynamic+*reweighting* strategy outperforms the unweighted dynamic variant, demonstrating the effectiveness of reweighting. Nevertheless, the uniform schedule also attains small regret, which we attribute to the simplicity of our threshold-expert class.

## B   Proof of Lemma 3

Denote $\ell_s(\cdot,\cdot) := \alpha_s \cdot \ell(\cdot,\cdot)$ and $K := |\mathcal{H}|$ for notation convenience. Let $W^s = \sum_{k=1}^{K} \mathbf{w}^s[k]$ be the potential. We have:

$$
\begin{aligned}
\log \frac{W^s}{W^{s-1}} &= \log \sum_{k=1}^{K} \frac{\mathbf{w}^{s-1}[k]}{W^{s-1}} e^{-\eta \cdot \ell_s(h_k(X_{t_s}^{(s)}), Y_{t_s}^{(s)})} \\
&\leq -\eta \sum_{k=1}^{K} \frac{\mathbf{w}^{s-1}[k]}{W^{s-1}} \cdot \ell_s(h_k(X_{t_s}^{(s)}), Y_{t_s}) + \frac{\eta^2}{8} \\
&\leq -\eta \cdot \ell_s \left( \sum_{k=1}^{K} \frac{\mathbf{w}^{s-1}[k]}{W^{s-1}} \cdot h_k(X_{t_s}^{(s)}), Y_{t_s}^{(s)} \right) + \frac{\eta^2}{8} \\
&= -\eta \ell_s(\hat{h}_{s-1}(X_{t_s}^{(s)}), Y_{t_s}^{(s)}) + \frac{\eta^2}{8},
\end{aligned}
$$

where the first inequality follows by Hoeffding's lemma [4, Lemma A.1] and the fact that $\alpha_s \in [0,1]$, the second inequality follows by Jensen's inequality and convexity of $\ell_s$, and the last identity follows

by the definition of $\hat{h}_{s-1}$ (see Algorithm 3). Summing from $s = 1$ to $S$, we have:

$$\log \frac{W^S}{W^1} \leq -\eta \sum_{s=1}^{S} \ell_s(\hat{h}_{s-1}(X_{t_s}^{(s)}), Y_{t_s}^{(s)}) + \frac{\eta^2 T}{8}.$$

The lemma follows from the fact that:

$$\log W^S \geq -\eta \inf_{k \in [K]} \sum_{s=1}^{S} \ell_s(h_k(X_{t_s}^{(s)}), Y_{t_s}^{(s)}), \qquad \log W^1 = \log K,$$

and setting $\eta = \sqrt{(\log K)/S}$. $\qquad \square$

**Remark 3.** *Note that the main difference compared to classical regret analysis of the EWA algorithm [4, Theorem 2.2] is that the losses $\ell_s$ at different time steps $s$ may vary.*

## C  Proof of Theorem 2

By [1, Lemma 12], the class $\mathcal{H}$ admits a *finite* expert class $\mathcal{G} \subset \mathcal{X}^* \to \{0, 1\}$ such that for any $h \in \mathcal{H}$ and data $\{Z_s^{(s)} := (X_{t_s}^{(s)}, Y_{t_s}^{(s)})\}_{s \in [S]}$, there exists $g \in \mathcal{G}$ satisfying:

$$\forall s \in [S], \quad h(X_{t_s}^{(s)}) = g(\{X_{t_i}^{(i)}\}_{i \leq s-1}, X_{t_s}^{(s)}).$$

This implies that, when running Algorithm 3 over $\mathcal{G}$, the incurred (weighted) regret against $\mathcal{H}$ is always upper bounded by the regret against $\mathcal{G}$. Moreover, by [1], we have:

$$\log |\mathcal{G}| \leq O\left(\mathsf{Ldim}(\mathcal{H}) \cdot \log S\right).$$

Invoking Lemma 3 on expert class $\mathcal{G}$, the (weighted) regret for $\mathcal{H}$ on the *queried* examples grows as:

$$O(\sqrt{S \log |\mathcal{G}|}) \leq O(\sqrt{S \log S \cdot \mathsf{Ldim}(\mathcal{H})})$$

The result then follows from the same arguments as in Lemma 4. $\qquad \square$

## D  Proof of Theorem 3

This appendix presents the complete proof of Theorem 3. Before showing the formal proof, we distinguish between the following two settings for *query strategies*:

- **Oblivious Query Strategy**: A query strategy that is *independent* of the data;
- **Adaptive Query Strategy**: A query strategy that *depends* on the data.

Similarly, we distinguish between the following two settings for *Nature*:

- **Oblivious Nature**: An adversarial environment where the data-generating process is chosen *independently* of the learner's actions;
- **Adaptive Nature**: An adversarial environment where the data-generating process is chosen *dependently* on the learner's actions.

This classification yields four lower-bound regimes—(O-O), (O-A), (A-O), and (A-A)—where the first letter specifies the query strategy and the second the behavior of Nature. Formally, a label (X-Y) asserts that *for every* query strategy of type X, *there exists* a data-generation process of type Y that achieves the stated lower bound.

The following proposition is straightforward:

**Proposition 2.** *The following implications hold among the different types of lower bounds:*

$$(A\text{-}O) \Rightarrow \{(O\text{-}O), (O\text{-}A), (A\text{-}A)\}$$

*Proof.* The proof follows from the facts that (A-X) $\Rightarrow$ (O-X) and (X-O) $\Rightarrow$ (X-A) for all X $\in \{O, A\}$. The first implication holds because a lower bound that applies to all Adaptive query strategies also applies to all Oblivious strategies under the same type X of data process. The second implication holds because an Oblivious Nature is a special case of an Adaptive Nature. $\qquad \square$

Proposition 2 essentially states that if a lower bound holds for any adaptive query strategy with an oblivious data process, then it is the *strongest*, as it also holds in the other three settings. Observe that the lower bounds we established in the proof of Section 3.2 can be classified as type (O-A), which is the weakest type among our classification above.

We now present below the complete proof of Theorem 3 with the *strongest* type (A-O). For the reader's convenience, we restate the theorem below:

**Theorem 4.** *Let $\mathcal{H} \subset \{0,1\}^{\mathcal{X}}$ be a binary-valued class of finite Littlestone dimension. Then, for any learning algorithm $\mathcal{A}$ with* adaptive *query strategy and a query budget $S$, there exists an* oblivious *data process $\mathbf{Z}$ such that:*

$$\mathrm{risk}_T^o(\mathcal{A}, \mathbf{Z}) - \inf_{h \in \mathcal{H}} \mathrm{risk}_T^o(h, \mathbf{Z}) \geq \Omega\left(\frac{T \cdot \sqrt{\mathsf{Ldim}(\mathcal{H})}}{\sqrt{S}}\right).$$

The proof of Theorem 4 relies on several technical lemmas, as follows.

**Lemma 6.** *Let $B_1, \ldots, B_m \overset{i.i.d.}{\sim} \mathrm{Bernoulli}\left(\frac{1}{2}\right)$. Choose indices $I_1, \ldots, I_k$ independently and uniformly from $\{1, \ldots, m\}$ (with replacement) and set $Y_t := B_{I_t}$ for $t = 1, \ldots, k$. Let $J$ be uniformly sampled from $\{1, \ldots, m\}$, independently of all previous choices. Let $S \subseteq \{I_1, \ldots, I_k\}$ be the set of all* distinct *sampled indices and denote $\tilde{k} = |S|$. Define*

$$\bar{Y}_{\mathrm{distinct}} := \frac{1}{\tilde{k}} \sum_{j \in S} B_j = \frac{1}{\tilde{k}} \sum_{j \in S} Y_{t(j)},$$

*where for each $j \in S$, $t(j)$ is any index with $I_{t(j)} = j$. Then*

$$\mathbb{E}[B_J \mid Y_1, \ldots, Y_k] = \frac{1}{2} + \frac{\tilde{k}}{m}\left(\bar{Y}_{\mathrm{distinct}} - \frac{1}{2}\right).$$

*Proof.* Condition on $(Y_1, \ldots, Y_k)$. For each $j \in S$, $B_j$ is revealed exactly as $Y_{t(j)}$, hence $\mathbb{E}[B_j \mid Y_1, \ldots, Y_k] = Y_{t(j)}$. For $j \notin S$, $B_j$ remains independent $\mathrm{Bernoulli}\left(\frac{1}{2}\right)$, so $\mathbb{E}[B_j \mid Y_1, \ldots, Y_k] = \frac{1}{2}$.

Averaging over the uniform $J$,

$$\mathbb{E}[B_J \mid Y_1, \ldots, Y_k] = \frac{1}{m} \sum_{j=1}^{m} \mathbb{E}[B_j \mid Y_1, \ldots, Y_k] = \frac{1}{m}\left(\sum_{j \in S} B_j + \sum_{j \notin S} \frac{1}{2}\right).$$

Simplifying,

$$\frac{1}{m}\left(\tilde{k}\, \bar{Y}_{\mathrm{distinct}} + (m - \tilde{k}) \cdot \frac{1}{2}\right) = \frac{1}{2} + \frac{\tilde{k}}{m}\left(\bar{Y}_{\mathrm{distinct}} - \frac{1}{2}\right),$$

as claimed. $\qquad \square$

**Lemma 7.** *Let $\Phi : \{0,1\}^k \to [0,1]$ be any (possibly random) predictor. Then, with the notation as in Lemma 6, we have:*

$$\mathbb{E}\left[|\Phi(Y_1, \cdots, Y_k) - B_J| \mid Y_1, \cdots, Y_k\right] \geq \frac{1}{2} - \left|\frac{\tilde{k}}{m}\left(\bar{Y}_{distinct} - \frac{1}{2}\right)\right|.$$

*Proof.* Condition on $Y_1, \cdots, Y_k$, we denote $p := \mathbb{E}[B_J \mid Y_1, \cdots, Y_k]$. Then, for any given $b \in [0,1]$, we have

$$\begin{aligned}
\mathbb{E}\left[|b - B_J| \mid Y_1, \cdots, Y_k\right] &= (1-p) \cdot b + p \cdot (1 - b) \\
&\geq \min\{p, 1 - p\} \\
&= \min\left\{\frac{1}{2} + \frac{\tilde{k}}{m}\left(\bar{Y}_{\mathrm{distinct}} - \frac{1}{2}\right), \frac{1}{2} - \frac{\tilde{k}}{m}\left(\bar{Y}_{\mathrm{distinct}} - \frac{1}{2}\right)\right\} \\
&= \frac{1}{2} - \left|\frac{\tilde{k}}{m}\left(\bar{Y}_{\mathrm{distinct}} - \frac{1}{2}\right)\right|,
\end{aligned}$$

where the second identity follows from Lemma 6. The lemma follows by taking expectation on both sides over the internal randomness of $\Phi$. $\qquad \square$

**Lemma 8.** *With the notations of Lemma 6, let $S \subseteq \{I_1, \ldots, I_k\}$, $\tilde{k} = |S|$, $\bar{Y}_{\text{distinct}} := \frac{1}{\tilde{k}} \sum_{j \in S} B_j$. Then, conditioning on the indices $I_1, \ldots, I_k$ (equivalently on $S$),*

$$\mathbb{E}\left[ \left| \frac{\tilde{k}}{m} \left( \bar{Y}_{\text{distinct}} - \tfrac{1}{2} \right) \right| \, \Big| \, I_1, \ldots, I_k \right] \leq \frac{\sqrt{\tilde{k}}}{2m}.$$

*Consequently,*

$$\mathbb{E}\left[ \left| \frac{\tilde{k}}{m} \left( \bar{Y}_{\text{distinct}} - \tfrac{1}{2} \right) \right| \right] \leq \frac{\mathbb{E}\left[ \sqrt{\tilde{k}} \right]}{2m} \leq \frac{\sqrt{\mathbb{E}\left[ \tilde{k} \right]}}{2m} = \frac{1}{2m} \sqrt{m \left( 1 - \left( 1 - \tfrac{1}{m} \right)^k \right)} \leq \frac{\sqrt{k}}{2m}.$$

*Proof.* Condition on $S$ (hence on $\tilde{k}$). Write $X_j := B_j - \frac{1}{2}$, so the $\{X_j\}_{j \in S}$ are independent, mean 0, and take values in $\{\pm \frac{1}{2}\}$. Then

$$\bar{Y}_{\text{distinct}} - \tfrac{1}{2} = \frac{1}{\tilde{k}} \sum_{j \in S} X_j, \qquad \left| \frac{\tilde{k}}{m} \left( \bar{Y}_{\text{distinct}} - \tfrac{1}{2} \right) \right| = \frac{1}{m} \left| \sum_{j \in S} X_j \right|.$$

By the Khintchine's inequality (upper bound in Lemma 5),

$$\mathbb{E}\left[ \left| \sum_{j \in S} X_j \right| \, \Big| \, S \right] \leq \sqrt{\mathbb{E}\left[ \left( \sum_{j \in S} X_j \right)^2 \, \Big| \, S \right]} = \sqrt{\sum_{j \in S} \mathbb{E}[X_j^2]} = \frac{1}{2} \sqrt{\tilde{k}}.$$

Dividing by $m$ yields the conditional bound $\mathbb{E}\left[ \left| \frac{\tilde{k}}{m} \left( \bar{Y}_{\text{distinct}} - \tfrac{1}{2} \right) \right| \, \Big| \, S \right] \leq \frac{\sqrt{\tilde{k}}}{2m}$. The second bound follows from the concavity of the square root: $\mathbb{E}[\sqrt{\tilde{k}}] \leq \sqrt{\mathbb{E}[\tilde{k}]}$. Finally, for sampling $k$ times with replacement from $m$ bins, $\mathbb{E}[\tilde{k}] = m \left( 1 - \left( 1 - \frac{1}{m} \right)^k \right)$, giving the stated closed form. $\square$

*Proof of Theorem 4.* We start by considering the simplified case where $\mathcal{H} := \{h_0, h_1\}$ with $h_0(\mathbf{x}) = 0$ and $h_1(\mathbf{x}) = 1$ for some fixed $\mathbf{x}$.

We partition the time horizon into $T \cdot S$ (small) blocks each of length $\frac{1}{S}$. We will assign the same label within each block, and assign the feature $\mathbf{x}$ for all time steps. Let

$$B_1, \cdots, B_m \overset{i.i.d.}{\sim} \text{Bernoulli}\left( \frac{1}{2} \right)$$

for some $m$ to be determined. For each block $t$, we assign its label $Y_t$ in the following manner: we first sample $I_t \sim \text{Unif}(\{1, \cdots, m\})$ independent of all other randomness and set $Y_t := B_{I_t}$.

Let $t_1, \cdots, t_S$ be the blocks that are queried by any learning algorithm, possibly dependent on the data. We may assume, w.l.o.g., that all the queries are in different blocks (since the same queries within one block reveal no additional information as the labels are the same). Now, consider any block $t$ between $t_s$ and $t_{s+1}$ that is *not* queried. By the law of total probability, we have:

$$\mathbb{E}\left[ |\hat{h}_s(\mathbf{x}) - Y_t| \right] = \mathbb{E}\left[ \mathbb{E}\left[ |\hat{h}_s(\mathbf{x}) - Y_t| \mid Y_{t_1}, \cdots, Y_{t_S} \right] \right]$$

$$\geq \mathbb{E}\left[ \inf_{\Phi} \mathbb{E}\left[ |\Phi(Y_{t_1}, \cdots, Y_{t_S}) - Y_t| \mid Y_{t_1}, \cdots, Y_{t_S} \right] \right]$$

$$\geq \mathbb{E}\left[ \frac{1}{2} - \frac{\tilde{S}}{m} \left| \bar{Y}_{\text{distinct}} - \frac{1}{2} \right| \right], \quad \text{by Lemma 7}$$

$$\geq \frac{1}{2} - \frac{\sqrt{S}}{2m}, \quad \text{by Lemma 8}$$

where we have used the fact that the distribution of $Y_t$ conditioned on the queried observations $Y_{t_1}, \cdots, Y_{t_S}$ is exactly the same as $B_J$ in Lemma 6 for $k = S$. Since there are $(T \cdot S - S)$ blocks that are not queried and each has size $\frac{1}{S}$, the total risk is lower bounded by (for $S \leq m$) [4]:

$$\frac{1}{S} \cdot (T \cdot S - S) \cdot \left( \frac{1}{2} - \frac{\sqrt{S}}{2m} \right) \geq \frac{T}{2} - \frac{T\sqrt{S}}{2m} - \frac{1}{2}. \tag{8}$$

We now analyze the comparator loss. Denote by $T_j$ the total length of all blocks that are assigned with variable $B_j$ for $j \in [m]$, where $\sum_{j=1}^m T_j = T$. Conditioned on any realization of $T_j$'s, the comparator risk can be expressed as:

$$\mathbb{E}\left[ \min_{b \in \{0,1\}} \left\{ \sum_{j=1}^m T_j \cdot |b - B_j| \right\} \right].$$

Using a similar argument for proving (7), we conclude that:

$$\mathbb{E}\left[ \min_{b \in \{0,1\}} \left\{ \sum_{j=1}^m T_j \cdot |b - B_j| \right\} \right] \leq \frac{T}{2} - \frac{T}{2\sqrt{2m}}. \tag{9}$$

Putting everything together and taking $m := 5S$, we arrive at the *expected* regret lower bound:

$$\frac{T}{2\sqrt{2m}} - \frac{T\sqrt{S}}{2m} - \frac{1}{2} \geq \frac{0.123 \cdot T}{\sqrt{S}} - \frac{1}{2} \geq \Omega\left( \frac{T}{\sqrt{S}} \right).$$

The lower bound for the simplified case now follows by choosing the *oblivious* data process to be the (deterministic) realization of the $Y_t$'s that attains the expected regret lower bound above, which must exist by the probabilistic method.

For general class $\mathcal{H}$ with Littlestone dimension $d := \mathsf{Ldim}(\mathcal{H})$, we construct the following *oblivious* data process. We partition the time horizon $T$ into $d$ phases, each of length $T/d$. Let $\tau$ be a Littlestone tree of $\mathcal{H}$ of depth $d$ (see [1, Lemma 14]), and let $\mathbf{x}_1$ be the root of $\tau$. We assign the feature $\mathbf{x}_1$ during the first phase and choose labels as in the simplified case above [5], for some value $m$ to be determined. Based on the realization of labels in phase one, we set the feature $\mathbf{x}_2$ for phase two to be the child of $\mathbf{x}_1$ in $\tau$ whose outgoing edge label minimizes the comparator loss, and generate labels in the same way using *fresh* randomness. We repeat this process for all $d$ phases.

By the definition of Littlestone tree, there must exist some $h \in \mathcal{H}$ that attains the minimum comparator loss in each of the $d$ phases. Invoking (9), the total comparator loss is upper bounded by

$$\sum_{r=1}^d \left( \frac{T/d}{2} + \frac{T/d}{2\sqrt{2m}} \right) = \frac{T}{2} - \frac{T}{2\sqrt{2m}}.$$

We now lower bound the risk of the learning algorithm. To avoid overly technical complications, we assume that the learner assigns a fixed (but arbitrary) query budget $S_r$ in phase $r \in [d]$, although the actual query times can be chosen completely arbitrarily [6]. Note that in this case, the learner may assign many queries in a single phase, so $S_r$ may exceed $m$ for some $r$. Fortunately, the upper bound in Lemma 8 holds even if $k \geq m$. Setting $k := S_r$ for all $r \in [d]$ and invoking a similar argument as in (8), we obtain the following lower bound on the learner's total risk:

$$\sum_{r=1}^d \frac{T/d}{2} - \frac{T}{d}\left( \frac{\sqrt{S_r}}{2m} + \frac{S_r}{2m\sqrt{m}} \right) - O\left( \frac{S_r}{S} \right) = \frac{T}{2} - \frac{T \cdot S}{2dm\sqrt{m}} - \frac{T}{2dm}\left( \sum_{r=1}^d \sqrt{S_r} \right) - O(1)$$

$$\geq \frac{T}{2} - \frac{T \cdot S}{2dm\sqrt{m}} - \frac{T\sqrt{dS}}{2dm} - O(1),$$

---

[4]For $S \geq T^2$, one may choose the block length as $\frac{1}{S^2}$ to get rid of the $\frac{1}{2}$ additive term.

[5]We still set the block length to $\frac{1}{S}$, where $S$ is the total query budget.

[6]The fully adaptive queries can be handled via a high-probability version of Lemma 8 using Bernstein's inequality (with additional log factors). We omit the details, as they are technical and not central to this work.

where we used the fact that $\sum_{r=1}^{d} S_r \leq S$, and the final inequality follows from Cauchy–Schwarz. Therefore, by setting $m := 20S/d$, the total expected regret is lower bounded by

$$\frac{T}{2\sqrt{2m}} - \frac{T \cdot S}{2dm\sqrt{m}} - \frac{T\sqrt{dS}}{2dm} - O(1) \;\geq\; \frac{0.048 \cdot T}{\sqrt{S/d}} - O(1) \;\geq\; \Omega\left(\frac{T \cdot \sqrt{d}}{\sqrt{S}}\right).$$

This completes the proof of Theorem 4. $\qquad\qquad\qquad\qquad\qquad\qquad\qquad\qquad\qquad\square$

**Remark 4.** *Note that a key technical innovation in our proof—compared to classical lower bound constructions such as in [1]—is that the expected risk incurred by the learning algorithm is no longer $T/2$. Instead, in our construction, the learner's risk and that of the comparator are subtly linked through the choice of $m$. The final regret bound leverages the crucial fact that, as $m$ increases, the learner's risk grows faster than the comparator's loss, enabling a non-trivial lower bound on regret via appropriate tuning of $m$.*

To illustrate the necessity of our technical argument in the proof of Theorem 4, we explain why a naive approach—similar to the one used in [1]—fails to yield the optimal lower bound. Suppose we partition the time horizon $T$ into blocks of length $\Delta$, and assign the label uniformly at random from $\{0, 1\}$ within each block. In this setting, the learner's expected risk is *not* $T/2$. Instead, the learner can query at the beginning of each block and repeat the observed label for the entire block, incurring zero risk on any queried block. Therefore, the total expected risk becomes $\frac{T-S\cdot\Delta}{2}$. On the other hand, a standard argument shows that the comparator loss (for the class $\{h_0, h_1\}$) is upper bounded by $\frac{T}{2} - O(\sqrt{\Delta \cdot T})$. To obtain non-trivial regret, one must have

$$\sqrt{\Delta \cdot T} \geq S \cdot \Delta.$$

This implies $\Delta \leq \frac{T}{S^2}$, and the resulting regret lower bound is $\Omega\left(\frac{T}{S}\right)$, which is significantly weaker than the optimal $\Omega\left(\frac{T}{\sqrt{S}}\right)$ regret.

