# OpenReview forum: "Agnostic Continuous-Time Online Learning"
_NeurIPS.cc/2025/Conference — NeurIPS 2025 poster_

### Official Review · Reviewer_aPug · 2025-06-15

**Clarity:** 3
**Significance:** 4
**Originality:** 4
**Rating:** 5
**Confidence:** 4

**Summary:**

This paper investigates agnostic continuous-time online learning. In contrast to the classical formulation of online learning, in which the input is a discrete sequence, the input here is a continuous data stream. While previous research has addressed the realizable setting, the more realistic and challenging agnostic setting has remained largely unexplored.  The authors demonstrate that, given a hypothesis class with Littlestone dimension $d$, the regret of online learning this class is exactly $\tilde{\Theta}(T\sqrt{d / S})$ for both oblivious and adaptive adversarial settings, where $T$ represents the time-span and $S$ denotes the number of queries the algorithm can make.

**Questions:**

1. I am concerned that Lemma 6 may not hold as $I_1,\dots,I_k$ could overlap. For example, let $m = k = 2$. Given that $Y_1 = Y_2 = 0$, it holds that $E[B_J \mid Y_1=Y_2=0]>0$ since it is possible (with positive probability) that $J\neq I_1 = I_2$ and $B_J = 1$. However, the Lemma gives $1/2 + k/m(\bar{Y}  - 1/2) = 0$.  As this lemma serves as a key inequality in the proof of the lower bound, can you provide a more detailed explanation regarding its validity?
2. It seems the lower bound only holds when $S$ is not excessively large: in the formula before line 413, the inequality requires $T / \sqrt{S} = \Omega(1)$ (i.e., $S = O(T^2)$). In contrast, the upper bound seems to hold even as $S \to \infty$. Therefore, the bounds do not match in the extreme regime. Is my understanding correct?

**Ethical Concerns:**

["NO or VERY MINOR ethics concerns only"]

**Final Justification:**

This paper investigates agnostic continuous-time online learning and provides tight upper and lower bounds. Though there were some issues in the original proof, they were fully addressed in the rebuttal. I recommend acceptance.

**Limitations:**

yes

**Quality:**

4

**Strengths And Weaknesses:**

**Strengths:**

The authors establish a nearly tight $\tilde{\Theta}(T\sqrt{d / S})$ regret bound for the fundamental problem of agnostic continuous-time online learning. For an oblivious adversary, they propose a simple black-box reduction to the discrete case. In the case of an adaptive adversary, they present a novel re-weighting scheme attaining the same bound. Additionally, they provide a matching lower bound that holds even for the most challenging scenario (an adaptive query strategy against an oblivious nature).

Furthermore, the organization and presentation of the paper are quite commendable, making it easy to read and understand.

**Weakness:**

I did not identify any significant weakness. However, I have concerns regarding the correctness of Lemma 6. Please refer to "Questions" for further details.

In the preliminaries, the authors define the protocol (Protocol 1) in a way that it chooses a predictor $\hat{h}_s\in\mathcal{Y}^{\mathcal{X}}$. However, it appears that sometimes the output is indeed a distribution over predictors (particularly in the adaptive setting, where this is necessary), and at any time step the actual predictor is drawn from this distribution (rather than keep outputting $\hat{h}_s$ for a period). I recommend the authors clarify this part to make the definition consistent.

---

> ### Author Rebuttal · Authors · 2025-07-24
>
> We thank the reviewer for the thoughtful feedback and helpful comments. We address the questions raised below:
>
> **Q1:** This is indeed a bug, and we are grateful to the reviewer for identifying it. The correct expression should be
>
> $$
> \frac{1}{2} + \frac{\tilde{k}}{m} \left( \bar{Y}_{\mathrm{distinct}} - \frac{1}{2} \right),
> $$
>
> where $\tilde{k} \le k$ is the number of *distinct* $I_t$'s, and $\bar{Y}_{\mathrm{distinct}}$ is the average of all $Y_t$'s corresponding to distinct indices $I_t$. The same replacement should also be made in Lemmas 7 and 8.
> This correction does not affect the subsequent argument, as the upper bound in Lemma 8 still holds since $\tilde{k} \le k$. We can apply Khintchine's inequality (upper bound) to obtain
>
> $$
> \frac{\tilde{k}}{m} \left| \bar{Y}_{\mathrm{distinct}} - \frac{1}{2} \right| \le \frac{\sqrt{\tilde{k}}}{2m} \le \frac{\sqrt{k}}{2m},
> $$
>
> by conditioning on the set of distinct indices. This provides an even better constant $\frac{1}{2}$ compared to $\frac{\sqrt{2}}{2}$, as the counters $N_j$ are simply indicators. We will make this correction in the revised version and again thank the reviewer for the careful reading.
>
> **Q2:** Indeed, our current lower bound proof is only non-vacuous when $S = O(T^2)$, due to the constant factor in (8). For general $S \ge T^2$, the $\Omega(T/\sqrt{S})$ lower bound still holds. This can be remedied by selecting the block length to be $1/S^2$ (instead of $1/S$ as in the paper), so that the constant factor in (8) becomes $O(1/S)$, which is negligible compared to the main term $T/\sqrt{S}$. We will clarify this in the revision.
>
> **Randomness of Predictors:** Indeed, for the 0–1 loss, it is necessary for the predictor to be randomized in order to achieve sublinear regret in the adaptive case. We will clarify in Protocol 1 that the predictor $\hat{h}_s$ is allowed to use internal randomness.

---

> > ### Comment · Reviewer_aPug · 2025-08-02
> >
> > Thank you for your response. After reviewing the rebuttal, I believe the issues I raised in my review have been adequately addressed. I will maintain my positive score and recommend acceptance.

---

### Official Review · Reviewer_ho6c · 2025-06-26

**Clarity:** 3
**Significance:** 3
**Originality:** 2
**Rating:** 4
**Confidence:** 4

**Summary:**

This paper considers a continuous online learning framework in which a learner selects discrete query times at which the data stream is observed (to evaluate the loss) and the learning strategy is updated. This framework was recently studied in [7], but only in the realizable setting—where the data is assumed to be generated by a function within the hypothesis class. In contrast, the present work focuses on the more challenging agnostic setting. The authors generalize the notion of expected regret to this continuous-time framework by integrating the loss over time, rather than summing it at discrete points.

Two settings are considered: the oblivious data stream, where the data is generated arbitrarily in advance and independently of the learner’s randomness; and the adaptive data stream, where the data stream may depend on any past information. For both settings, the authors propose a black-box reduction to an expert algorithm, using the observed losses at the random query times as unbiased estimators of the integrated continuous loss. The regret bound then follows by plugging these estimators into the standard regret analysis of the expert algorithm.

In addition to upper bounds, the authors provide a nearly matching lower bound (up to logarithmic factors) in the oblivious setting, showing that the regret guarantees are essentially optimal.

**Questions:**

- Is it possible to derive high-probability guarantees, rather than bounds only in expectation? Alternatively, can one obtain upper bounds on the actual expected regret (i.e., using the infimum over $h$ in Equation (5)) instead of pseudo-regret bounds (i.e., the expectation as in Equation (4))?

- The results are presented differently for the oblivious and adaptive settings. Is it not possible to unify the presentation, perhaps by giving a generic analogue of the black-box reduction in Theorem 1 for the adaptive setting—similar to what is done for the oblivious case—rather than relying on the more case-specific Lemma 4 and Theorem 2? The motivation for diverging in presentation is not entirely clear.

**Ethical Concerns:**

["NO or VERY MINOR ethics concerns only"]

**Final Justification:**

I have read other reviews and the rebuttal. I think the authors address well my concerns. In particular, they added synthetic experiments that seem to well illustrate their theoretical results. I lean toward acceptance.

**Limitations:**

Yes

**Quality:**

3

**Strengths And Weaknesses:**

Strenghts:
- The paper is well-written and easy to follow.
- The problem considered is interesting and the authors provide a clean and complete analysis with tight upper and lower bounds.

Weaknesses:
- The results are restricted to finite hypothesis classes. Is it not possible to extend the black-box reduction to continuous hypothesis classes, and more generally to reduce to any discrete online convex optimization algorithm, rather than limiting to expert algorithms?
- The paper would benefit from synthetic numerical experiments to illustrate the practical implications of the continuous-time framework on regret. In particular, when setting $T = S$, we would expect to recover similar performance—does this hold in practice? Additionally, how stable are the results, given that they rely on an estimation technique?
- The proofs appear to be fairly standard and rely on combining known results: the agnostic lower bound follows techniques from [1]; the sampling process for query points is borrowed from [7]; and the use of unbiased loss estimators within an expert algorithm mirrors approaches commonly used in adversarial bandits and online learning with partial feedback.

---

> ### Author Rebuttal · Authors · 2025-07-24
>
> We thank the reviewer for the thoughtful feedback and helpful comments. We address the questions raised below:
>
> **W1:** In fact, our results for the adaptive case also apply to hypothesis classes with finite Littlestone dimension, not just finite classes, as shown in Theorem 2. For general OCO algorithms, our reduction could still apply by running the algorithm (e.g., OGD or FTRL) on the *weighted* loss function. We only require a regret guarantee of the form stated in Lemma 3. We thank the reviewer for pointing this out and will be happy to include this clarification in the revised version.
>
> **W2 & Q1:** Our regret guarantee relies on estimating the loss, which is bounded by $\Delta \approx T/S$. For $S = T$, the variance is constant; in this case, an application of Freedman's inequality for martingales (with filtration induced by $t_s$) yields a high-probability bound on the *learner's risk* (or any fixed comparator risk) with deviation $O(\sqrt{S}) = O(\sqrt{T})$. For general $S$, the deviation scales as $O(\Delta \sqrt{S}) = O(T/\sqrt{S})$, which matches the order of our expected regret $\tilde{O}(T\sqrt{\mathsf{Ldim}(\mathcal{H})/S})$.
>
> In the *oblivious* case, the learner's high-probability risk can be translated into a high-probability bound for the actual (pointwise) regret. However, for the *adaptive* case, it is unclear whether such a high-probability pointwise regret bound is attainable, since the optimal comparator may depend on the learner’s internal randomness. To our knowledge, no such *adaptive* high-probability pointwise regret bound exists even in the adversarial bandit setting. Nevertheless, the above argument still yields the following weaker form of adaptive high-probability regret: for any given expert $h \in \mathcal{H}$, with high probability, the (pointwise) regret of the learner against $h$ is $\tilde{O}(T\sqrt{\mathsf{Ldim}(\mathcal{H})/S})$.
>
> **W3:** We agree that many of our technical ingredients are inspired by classical ideas. However, synthesizing these ideas in our (novel) setting is far from straightforward. For example, the selection of the weighting factor is derived from the randomness of the query times $t_s$ and is not arbitrarily chosen. The fact that it may appear natural in hindsight does not imply that it is easy to derive or even to recognize it's existence.
> Moreover, our lower bound proof in Appendix D significantly extends the construction in Ben-David et al. [1] by designing an *oblivious* process (see Remark 4 and the accompanying discussion).
>
> **Q2:** We thank the reviewer for the helpful suggestion. As noted in our response to **W1**, a general reduction is indeed achievable in the OCO framework. We will include this discussion in the main text following Theorem 2.
>
> **Synthetic Experiments:**
> We conducted synthetic experiments to demonstrate the robustness of our algorithm. We used the expert class
>
> $$
> \mathcal{H} = \\{ h_{\theta} := \mathbf{1}\\{x \le \theta\\} : \theta \in \\{0, 0.001, \ldots, 1.000\\} \\},
> $$
>
> so that $|\mathcal{H}| = 1000$. We set the time horizon to $T = 5$ and discretize it into 1000 small blocks to approximate the integral regret *without* normalization. So, the simulated regret values reported below are scaled by a factor of $200\times$ compared to the actual continuous-time regret used in the paper. We simulate our **Algorithm 1** under different query budgets $S$, using the Exponential Weights Algorithm (EWA) as the expert algorithm $\mathcal{B}$, on the following data streams (all features are generated iid uniform from $[0,1]$):
> 1. **`alt_blocks`**: The stream is divided into $S$ equal chunks where the label for the chunks follows an (deterministic) alternating 0/1 pattern.
> 2. **`iid`**: Each label is drawn independently from a Bernolli source with parameter $p=0.3$.
> 3. **`realizable`**: The labels are generated by a ground-truth function from $\mathcal{H}$.
> 4. **upper bnd.**: Our theoretical upper bound.
> 5. **lower bnd.**: Our theoretical lower bound.
>
> The considered data processes span a broad spectrum, ranging from stationary to highly non-stationary regimes. Results are shown below (each experiment is repeated 100 times, and the average regret is reported):
>
> | $S$  | `alt_blocks` | `iid`   | `realizable` | upper bnd. |
> |------|--------------|---------|--------------|------------|
> | 5    | 161.98       | 172.93  | 197.13       | 1175.9     |
> | 10   | 102.30       | 138.96  | 144.92       | 831.4      |
> | 20   | 65.91        | 99.07   | 120.65       | 588.0      |
> | 40   | 57.16        | 79.34   | 86.10        | 415.6      |
> | 80   | 47.65        | 70.96   | 63.50        | 293.3      |
> | 160  | 35.97        | 61.76   | 48.97        | 207.6      |
> | 320  | 41.86        | 35.47   | 36.01        | 147.0      |
> | 640  | 42.18        | 27.73   | 22.81        | 103.9      |
>
> We also conducted simulations on the *hard* data stream `lower_stream` used in our lower bound proof (cf. Appendix D):
>
> | $S$  | lower bnd. | `lower_stream` | upper bnd. |
> |------|-------------|----------------|-------------|
> | 5    | 71.6        | 132.77         | 1175.9      |
> | 10   | 50.6        | 42.80          | 831.4       |
> | 20   | 35.8        | 89.83          | 588.0       |
> | 40   | 25.3        | 39.83          | 415.6       |
> | 80   | 17.9        | 29.51          | 293.3       |
> | 160  | 12.6        | 27.75          | 207.6       |
> | 320  | 8.9         | 28.40          | 147.0       |
> | 640  | 6.3         | 33.08          | 103.9       |
>
> **Observation 1:** For all four data streams, the regret is upper bounded by our theoretical upper bound. In fact, the curves have very similar slopes when plotted on a log-log scale, demonstrating that the $1/\sqrt{S}$ order is genuine.
>
> **Observation 2:** The regret for the `lower_stream` data is tightly sandwiched between our theoretical lower and upper bounds, demonstrating the optimality of our theoretical analysis.
>
> We expect similar results to hold for Algorithm 2 as well, since the weighting factor lies in $[0,1]$ and does not increase the variance of the loss estimation. We are happy to incorporate the experimental results reported above in our revision, along with additional discussion.

---

### Official Review · Reviewer_fUeB · 2025-06-29

**Clarity:** 3
**Significance:** 2
**Originality:** 3
**Rating:** 4
**Confidence:** 1

**Summary:**

This paper studies agnostic online learning from continueous-time data streams. learners have to interact with continually envolving data stream while making queries and updating model at strategically selected times. This paper develop theoretical framework for learning from both oblivious and adaptive data streams.

**Questions:**

1. The paper builds on and extends the update-and-deploy framework of Devulapalli and Hanneke [7], which focused on the realizable setting. Could the authors clearly delineate the conceptual and technical differences and challenges introduced by the agnostic setting?
2. The paper proposes a novel importance weighting scheme (Algorithm 3) to obtain unbiased loss estimates under adaptive query times with dynamic epochs. Could the authors clarify how sensitive is the performance and regret bound to the exact choice of weighting function?
3. Practical data streams can be complex and non-stationary beyond Littlestone dimension and scale characterization. Could the authors discuss or exemplify scenarios or hypothesis classes where these bounds translate to meaningful learning guarantees in real-world continuous data and elaborate on how these theoretical guarantees relate to computational or sample complexity in applications like high-frequency trading or personalized recommendations?

**Ethical Concerns:**

["NO or VERY MINOR ethics concerns only"]

**Limitations:**

Yes, the authors have acknowledged and discussed the limitations in Appendix A,

**Quality:**

2

**Strengths And Weaknesses:**

Strengths:
This paper develops generic algorithmic frameworks for agnostic continuous-time online learning under both oblivious and adaptive data streams with optimal regret guarantees, giving both upper and lower bounds.

Weakness:
No experimental results on either synthetic or real world data are conducted in the paper, it remains unclear whether the algorithms proposed can be used in practice.

---

> ### Author Rebuttal · Authors · 2025-07-24
>
> We thank the reviewer for the thoughtful feedback and helpful comments. We address the questions raised below:
>
> **Q1:** The primary conceptual difference compared to the framework of Devulapalli and Hanneke [7] is that our setting does *not* assume the existence of a perfect labeling rule from the hypothesis class. We also explicitly treat the query budget as a resource, which allows us to derive a precise quantitative trade-off between regret and the query budget.
>
> From a technical standpoint, the work in [7] applies dynamic querying without weighting, leveraging the fact that the plain loss serves as an overestimate that is sufficient to bound mistakes in the realizable setting. However, in the agnostic setting, such an overestimate is insufficient to lower bound the competitor's loss.  Our main technical innovation is the introduction of a weighting factor that produces exactly unbiased loss estimates, along with novel reductions to classic expert algorithms. Additionally, the lower-bounding techniques presented in Section 3 and Appendix D are entirely novel compared to those in [7].
>
> **Q2:** The weighting factor was not arbitrarily chosen, but rather derived from the randomness in the query times $t_s$, as shown in Lemma 2. In fact, other methods of selecting $t_s$ could also work, potentially leading to different weighting factors using the same derivation technique as in the proof of Lemma 2. Note that the weighting factor introduced in our Algorithm 3 lies in $[0,1]$, so it does not increase the variance of the estimator used in Algorithm 1 (which we simulate below).
>
> **Q3:** In fact, we consider the *worst-case* scenario; that is, our results apply to *any* non-stationary data-generating process. The "complexity" our framework addresses is the *query* (or update) budget. For example, in high-frequency trading, one must deploy a model that operates in real time, and updating the model can be costly—whether due to retraining overhead or concerns about system stability. As a result, updates must be sparse and strategic. Our work provides a worst-case performance baseline for such applications.
>
> **Synthetic Experiments:**
> We conducted synthetic experiments to demonstrate the robustness of our algorithm. We used the expert class
>
> $$
> \mathcal{H} = \\{ h_{\theta} := \mathbf{1}\\{x \le \theta\\} : \theta \in \\{0, 0.001, \ldots, 1.000\\} \\},
> $$
>
> so that $|\mathcal{H}| = 1000$. We set the time horizon to $T = 5$ and discretize it into 1000 small blocks to approximate the integral regret *without* normalization. So, the simulated regret values reported below are scaled by a factor of $200\times$ compared to the actual continuous-time regret used in the paper. We simulate our **Algorithm 1** under different query budgets $S$, using the Exponential Weights Algorithm (EWA) as the expert algorithm $\mathcal{B}$, on the following data streams (all features are generated iid uniform from $[0,1]$):
> 1. **`alt_blocks`**: The stream is divided into $S$ equal chunks where the label for the chunks follows an (deterministic) alternating 0/1 pattern.
> 2. **`iid`**: Each label is drawn independently from a Bernolli source with parameter $p=0.3$.
> 3. **`realizable`**: The labels are generated by a ground-truth function from $\mathcal{H}$.
> 4. **upper bnd.**: Our theoretical upper bound.
> 5. **lower bnd.**: Our theoretical lower bound.
>
> The considered data processes span a broad spectrum, ranging from stationary to highly non-stationary regimes. Results are shown below (each experiment is repeated 100 times, and the average regret is reported):
>
> | $S$  | `alt_blocks` | `iid`   | `realizable` | upper bnd. |
> |------|--------------|---------|--------------|------------|
> | 5    | 161.98       | 172.93  | 197.13       | 1175.9     |
> | 10   | 102.30       | 138.96  | 144.92       | 831.4      |
> | 20   | 65.91        | 99.07   | 120.65       | 588.0      |
> | 40   | 57.16        | 79.34   | 86.10        | 415.6      |
> | 80   | 47.65        | 70.96   | 63.50        | 293.3      |
> | 160  | 35.97        | 61.76   | 48.97        | 207.6      |
> | 320  | 41.86        | 35.47   | 36.01        | 147.0      |
> | 640  | 42.18        | 27.73   | 22.81        | 103.9      |
>
> We also conducted simulations on the *hard* data stream `lower_stream` used in our lower bound proof (cf. Appendix D):
>
> | $S$  | lower bnd. | `lower_stream` | upper bnd. |
> |------|-------------|----------------|-------------|
> | 5    | 71.6        | 132.77         | 1175.9      |
> | 10   | 50.6        | 42.80          | 831.4       |
> | 20   | 35.8        | 89.83          | 588.0       |
> | 40   | 25.3        | 39.83          | 415.6       |
> | 80   | 17.9        | 29.51          | 293.3       |
> | 160  | 12.6        | 27.75          | 207.6       |
> | 320  | 8.9         | 28.40          | 147.0       |
> | 640  | 6.3         | 33.08          | 103.9       |
>
> **Observation 1:** For all four data streams, the regret is upper bounded by our theoretical upper bound. In fact, the curves have very similar slopes when plotted on a log-log scale, demonstrating that the $1/\sqrt{S}$ order is genuine.
>
> **Observation 2:** The regret for the `lower_stream` data is tightly sandwiched between our theoretical lower and upper bounds, demonstrating the optimality of our theoretical analysis.
>
> We are happy to incorporate the experimental results reported above in our revision, along with additional discussion.

---

### Official Review · Reviewer_YgrP · 2025-07-02

**Clarity:** 3
**Significance:** 3
**Originality:** 2
**Rating:** 5
**Confidence:** 3

**Summary:**

This paper extends te setting of (realizable) continuous-time online learning from Devulapalli and Hanneke to the agnostic case. In this framework, there is a stream in continuous time of queries $X_t$, and the algorithm needs to make online predictions on the stream  by picking a hypothesis to use with a few discrete observation of the true labels $Y_t$ to the stream. The original assumed that the labels in the stream were generated by some hypothesis in the hypothesis class. This paper drops this assumption, and show algorithms (and some matching lower bounds) for both oblivious and adaptive adversaries. In the oblivious case the algorithm splits the time-horizon in equal length epochs and samples a uniform time for each epoch to see the label and use as a black-box an OL algorithm. In the adaptive case this may not work anymore, and they show how to use dynamic epochs, also used by Devulapalli and Hanneke, and dynamic step sizes on EWA to circumvent this issue.

**Questions:**

Small technical question, but DH24 assume the process $Z_t$  to be countable. I don't think this matters too much except on some of the technical conditions required later on (such as measurability of the sample paths). Do the authors think there are some technical kinks that might need to be ironed out because of this?

**Ethical Concerns:**

["NO or VERY MINOR ethics concerns only"]

**Final Justification:**

I did not have many concerns, and it seems like the other reviewers only raised minor points. I still lean towards accepting this submission.

**Limitations:**

The limitations of the setup seem to be well discussed in the paper.

**Paper Formatting Concerns:**

No formatting concerns

**Quality:**

3

**Strengths And Weaknesses:**

This paper studies a very natural (and more challenging) extension of the continuous-time online learning framework proposed by Devulapalli and Hanneke, which by itself it already a somewhat natural problem. Many of the techniques in the algorithms seem to drawn upon the ideas of DH24, but are certainly not trivial extensions in any way. The algorithm (or, better saying, reduction) for the oblivious case is clean and serves as a nice way to "warm-up" to the adaptive case, although the oblivious case is interesting in itself. Also, the slight modification of EWA  leads to a (relatively) elegant analysis. Unfortunately I did not have the time to check the proofs as I would like to have done, but the few derivations that I had time to check were relatively clearly exposed and seemed to be correct.

Overall, although there does not seem to be major novel ideas in the framework nor in the algorithms, this is a quite solid contribution, and shows optimal rates for both the oblivious and adaptive cases, which makes this a even more complete and solid work. While I have some questions, none of them seem to be crucial for clarity/significance/originality/quality of the paper.

---

> ### Author Rebuttal · Authors · 2025-07-24
>
> We thank the reviewer for the thoughtful feedback and helpful comments. Indeed, in our paper, we explicitly assume that the function induced by the loss is measurable for any sample path, so that the risk defined via integration is well-defined. The measurability and boundedness of $f(t)$ ensure that the function is absolutely integrable; thus, the exchange of the order of integration in Lemma 2 follows from Fubini’s theorem, which does not require assumption on the countability of the $Z_t$'s.
>
> From a practical point of view, one can also consider a sufficiently small partition of the time horizon to approximate the integral and thereby avoid measurability concerns—in which case, our result still holds.

---

> > ### Comment · Reviewer_YgrP · 2025-08-04
> >
> > I would like to thank the authors for their replies to my review and also to the other reviewers. I did not have many concerns, and it seems like the other reviewers only raised minor points. I still lean towards accepting this submission.

---

### Note · Authors · 2025-08-15

We thank all of the reviewers for their very positive assessment of our paper (initial scores of 5, 5, 4, 4). We comprehensively addressed all of the concerns raised by the reviewers to their satisfaction. To quote, "I did not have many concerns, and it seems like the other reviewers only raised minor points. I still lean towards accepting this submission." (YgrP), "I thank the authors for their rebuttal, which clarified several points." (ho6c), "After reviewing the rebuttal, I believe the issues I raised in my review have been adequately addressed." (aPug). We were not able to engage with reviewer FueB (initial score 4), who we still hope will be able to see our rebuttal and revise their score as appropriate.

We thank the reviewers for their constructive engagement and their strong assessment of our work.

---

### Decision · Program_Chairs · 2025-09-17

**Decision:**

Accept (poster)

**Comment:**

This paper provides the first guarantees for continuous-time online learning in the agnostic setting for hypothesis classes with a finite Littlestone dimension. All of the reviewers were positive about this submission and overall in favor of acceptance. For the camera-ready version, the authors need to include the fix to the bug in Lemma 6 that was discovered during the review process. Additionally, they are highly recommended to add the experimental results that they shared in the rebuttal, and provide context for the computational complexity and generality of the procedure that some reviewers believed was missing/unclear in the initial submission.